# THE HIDDEN LANGUAGE OF DIFFUSION MODELS

**Hila Chefer**[*1,2]     **Oran Lang**[1]     **Mor Geva**[3]     **Volodymyr Polosukhin**[1]     **Assaf Shocher**[1]
**Michal Irani**[1,4]     **Inbar Mosseri**[1]     **Lior Wolf**[2]

[1]Google Research     [2]Tel-Aviv University     [3]Google DeepMind     [4]Weizmann Institute

`https://hila-chefer.github.io/Conceptor/`

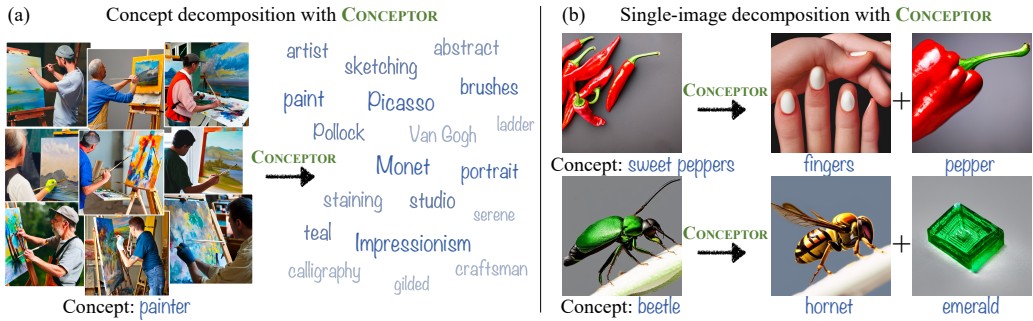

Figure 1: Concept interpretation with CONCEPTOR. (a) Given a set of representative concept images, CONCEPTOR learns to decompose the concept into a weighted combination of interpretable elements (font sizes indicate weights). The decomposition exposes interesting behaviors such as reliance on prominent painters and renowned artistic styles (*e.g.*, *"Monet"*, *"Impressionism"*). (b) Given a *specific* generated image, CONCEPTOR extracts its primary contributing elements, revealing surprising visual connections (*e.g.*, *"sweet peppers"* are linked to *"fingers"* due to their common shape).

## ABSTRACT

Text-to-image diffusion models have demonstrated an unparalleled ability to generate high-quality, diverse images from a textual prompt. However, the internal representations learned by these models remain an enigma. In this work, we present CONCEPTOR, a novel method to interpret the internal representation of a textual concept by a diffusion model. This interpretation is obtained by decomposing the concept into a small set of human-interpretable textual elements. Applied over the state-of-the-art Stable Diffusion model, CONCEPTOR reveals non-trivial structures in the representations of concepts. For example, we find surprising visual connections between concepts, that transcend their textual semantics. We additionally discover concepts that rely on mixtures of exemplars, biases, renowned artistic styles, or a simultaneous fusion of multiple meanings of the concept. Through a large battery of experiments, we demonstrate CONCEPTOR's ability to provide meaningful, robust, and faithful decompositions for a wide variety of abstract, concrete, and complex textual concepts, while allowing to naturally connect each decomposition element to its corresponding visual impact on the generated images.

## 1 INTRODUCTION

Generative models have demonstrated unprecedented capabilities to create high-quality, diverse imagery based on textual descriptions (Balaji et al., 2022; Gafni et al., 2022; Ramesh et al., 2021; Rombach et al., 2022; Saharia et al., 2022). While revolutionary, recent works have demonstrated that these models often suffer from heavy reliance on biases (Chuang et al., 2023; Luccioni et al., 2023) and occasionally also data memorization (Carlini et al., 2023; Somepalli et al., 2022). However, all these works draw conclusions by a simple external evaluation of the output images, while research on understanding the internal representations learned by the model remains scarce. Thus, our understanding of these impressive models remains limited.

---

*The first author performed this work as an intern at Google Research.

In this work, we aim to develop a method to interpret the inner representations of text-to-image diffusion models. Our approach draws inspiration from the field of concept-based interpretability (Li et al., 2021; Kim et al., 2017; Ghorbani et al., 2019), which proposes to interpret the model's decision-making for a given input by *decomposing* it into a set of elements that impact the prediction (*e.g.*, decomposing *"cat"* into *"whiskers"*, *"paws"*, *etc.*). Under this setting, an "interpretation" can be considered to be a *mapping function* from the internal state of the model to a set of concepts humans can understand (Kim et al., 2017). Notably, existing approaches for concept-based interpretability are not directly applicable to generative models, and, as we demonstrate, fall short of producing meaningful interpretations for such models. Therefore, we propose a novel method to produce concept-based explanations for diffusion models, which leverages their unique structure and properties.

Our method, CONCEPTOR, learns a *pseudo-token*, realized as a combination of interpretable textual elements (Fig. 1(a)). To obtain the pseudo-token, CONCEPTOR trains a neural network to map each word embedding in the vocabulary of the model to a corresponding coefficient, with the objective of denoising the concept images. The pseudo-token is then constructed as a linear combination of the top vocabulary elements weighted by their learned coefficients. This formulation allows us to exploit both the model's powerful ability to link between text and image, and the rich semantic information encapsulated in the text encoder. Through a large battery of experiments, we demonstrate that CONCEPTOR produces decompositions that are meaningful, robust, and faithful to the model.

We use CONCEPTOR to analyze the state-of-the-art text-to-image diffusion model, Stable Diffusion (Rombach et al., 2022) across various concepts, including concrete (*e.g.*, *"a nurse"*), abstract (*e.g.*, *"affection"*) and complex (*e.g.*, *"elegance on a plate"*) concepts, as well as homograph concepts (*e.g.*, *"a crane"*). CONCEPTOR reveals many interesting observations about the learned representations. (i) As demonstrated in Fig. 1(b), CONCEPTOR can be used to decompose a generated image to its own subset of driving elements. We find non-trivial combinations of features that control different visual aspects such as shapes, textures, and colors. (ii) We observe that some concepts such as *"a president"* or *"a rapper"* are represented mostly by *exemplars*, *i.e.*, well-known instances from the concept, such that the generated images are *interpolations* of those instances. (iii) We additionally find that consistent with previous work (Rassin et al., 2022), the model learns to mix the multiple meanings of homograph concepts. We expand those findings and discover cases where these meanings are leveraged *simultaneously*, creating images that mix both meanings in a *single object*. (iv) Finally, we demonstrate our method's effectiveness in the detection of non-trivial biases.

To conclude, our work makes the following contributions: (i) We present CONCEPTOR, a novel method to decompose a textual concept into a set of interpretable elements. Our method utilizes a unique form of decomposition in which a linear combination is learned as a mapping from the textual embedding space to a coefficient. (ii) We demonstrate profound learned connections between concepts that transcend textual correlations. (iii) We discover non-trivial structures in the learned decompositions such as interpolations of exemplars, reliance on renowned artistic styles, and mixing of different meanings of the concept. Finally, (iv) We demonstrate the detection of biases that are not easily observable visually. These observations can help discuss ethical questions on a factual basis.

## 2 RELATED WORK

**Text-guided image generation** Recently, impressive results were achieved for text-guided image generation with large-scale auto-regressive models (Ramesh et al., 2021; Yu et al., 2022) and diffusion models (Ramesh et al., 2022; Nichol et al., 2021; Rombach et al., 2022; Saharia et al., 2022). In the context of text-to-image diffusion models, a related line of work aims to introduce personalized concepts to a pre-trained text-to-image model by learning to map a set of images to a "token" in the text space of the model (Gal et al., 2022; Ruiz et al., 2022; Kumari et al., 2022). Importantly, these methods do not produce decomposable or interpretable information and mostly result in a rigid learned vector that resides outside of the model's distribution (Voynov et al., 2023).

**Concept-based interpretability** A similar analysis to ours was conducted on concept representations in the context of language models (Patel & Pavlick, 2022; Li et al., 2021; 2023b; Lovering & Pavlick, 2022), often through projections to the vocabulary space (Geva et al., 2022; 2023; Ram et al., 2022). For image classifiers based on CNNs, TCAV (Kim et al., 2017) propose the first concept-based explainability method by training a linear classifier over user-defined concepts. ACE (Ghorbani et al., 2019) leverages multi-resolution segmentation of the class images and clusters the bottleneck

representations of the crops. ICE (Zhang et al., 2020) and CRAFT (Fel et al., 2022) apply matrix factorization methods on a feature map matrix extracted from a set of patches to obtain the decomposition. Note that all these methods are not applicable directly to diffusion models. First, most of the methods are based on CNN architectures (Ghorbani et al., 2019) with non-negative activations (Zhang et al., 2020; Fel et al., 2022), thus cannot be directly generalized to negative feature matrices. Most importantly, all methods above perform concept importance estimation by measuring the shift in prediction or by employing saliency methods, both are not trivially applicable to a generative model.

**Object representation methods** A related line of work attempts to learn representations for images that hinge on the objects in the image. Singh et al. (2018) disentangle the background, shape, and appearance of the image for controllable generation. GIRAFFE (Niemeyer & Geiger, 2020) incorporates 3D scene representation to improve controllable image generation, Slot Attention (Locatello et al., 2020) and DTI Sprites (Monnier et al., 2021) provide improved decomposition with additional features, and GENESIS-V2 (Engelcke et al., 2021) learns object representations without iterative refinement. Note that all of the aforementioned methods only offer decomposition for a single image and, therefore fall short of providing concept-level explanations, which is our main objective. In the context of diffusion models, the closest work to ours is that of Liu et al. (2023), which proposes to use diffusion models for concept discovery. Importantly, all methods above provide representations that are based on concrete objects and parts that are visible in the image, thus are not applicable to our task of interpretability of internal representations (see Appendix G).

**Diffusion model interpretability** Shared text-image representations such as CLIP (Radford et al., 2021) have been analyzed in the past (Chefer et al., 2021a;b; Yun et al., 2023). However, none of these works has been generalized to generative models. As far as we can ascertain, the closest effort to explaining text-to-image models is a simple visualization of the cross-attention maps (Hertz et al., 2022; Han et al., 2023; Chefer et al., 2023). Some works (Rassin et al., 2022; Carlini et al., 2023) have attempted to investigate the images produced by text-to-image diffusion models, and have even found evidence of memorization (Carlini et al., 2023). However, these works rely entirely on the generated images, and are, therefore, time-consuming and require access to the data. For example Somepalli et al. (2022) were able to show memorization for less than $2\%$ of the generations with SD. Additionally, these works do not attempt to dissect the model's inner representations.

## 3 METHOD

**Preliminaries: Latent Diffusion Models (LDMs)** We apply our method over the state-of-the-art Stable Diffusion (SD) model (Rombach et al., 2022). SD employs a denoising diffusion probabilistic model (DDPM) (Sohl-Dickstein et al., 2015; Ho et al., 2020) over an input latent vector $z_T \sim \mathcal{N}(0,1)$ and gradually denoises it. Namely, at each timestep, $t = T, \ldots, 1$, the DDPM receives a noised latent vector $z_t$ and produces a less noisy vector $z_{t-1}$, which serves as the input to the next step.

During the denoising process, the model is typically conditioned on a text encoding for an input prompt $\mathcal{P}$, produced by a frozen CLIP text encoder (Radford et al., 2021), which we denote by $\mathcal{C}$. The text encoder converts the textual prompt $\mathcal{P}$ to a sequence of tokens, which can be words, sub-words, or punctuation marks. Then, the encoder's vocabulary, $\mathcal{V} \in \mathbb{R}^{N,d}$, is used to map each token in the prompt to an embedding vector $w \in \mathbb{R}^d$, where $d$ is the embedding dimension of the encoder, and $N$ is the number of tokens in the vocabulary. The DDPM model is trained to minimize the loss,

$$\mathcal{L}_{reconstruction} = \mathbb{E}_{z,\mathcal{P},\varepsilon \sim \mathcal{N}(0,1),t} \left[ ||\varepsilon - \varepsilon_\theta(z_t, t, \mathcal{C}(\mathcal{P}))||^2 \right], \text{for,} \tag{1}$$
$$z_t = \sqrt{\alpha_t} z + \sqrt{1 - \alpha_t} \varepsilon, \tag{2}$$

where $\varepsilon_\theta$ is a trained UNet (Ronneberger et al., 2015), and $0 = \alpha_T < \alpha_{T-1} < \cdots < \alpha_0 = 1$. In other words, during training, the input image $x$ is encoded to its corresponding latent vector $z$. A noise vector $\varepsilon$ and a timestep $t$ are drawn randomly. The noise vector $\varepsilon$ is then added to the latent vector $z$ as specified in Eq. 2, and the UNet is trained to predict the added noise $\varepsilon$.

**CONCEPTOR** Our goal is to interpret the internal representation of an input concept $c$ in a text-to-image diffusion model $\varepsilon_\theta$. Formally, given a prompt $\mathcal{P}^c$ for the concept $c$, we learn a decomposition for the concept using the vocabulary $\mathcal{V}$. This decomposition is realized as a pseudo-token $w^* \notin \mathcal{V}$ that is constructed as a weighted combination of a subset of tokens from $\mathcal{V}$, *i.e.*,

$$w^* = \sum_{i=1}^{n} \alpha_i w_i \qquad \text{s.t.} \qquad w_i \in \mathcal{V}, \alpha_1, \ldots, \alpha_n \geq 0 \tag{3}$$

where $n << N$ is a hyperparameter that determines the number of tokens to use in the combination.

Learning the set of $n$ vocabulary elements $w_i$ and their associated coefficients $\alpha_i$ is done separately for each concept $c$. We begin by collecting a training set $\mathcal{T}$ of $100$ concept images. These images provide the statistics for the concept features we wish to learn. Next, we construct our learned pseudo-token over the vocabulary $\mathcal{V}$. This construction serves two purposes. First, note that the vocabulary $\mathcal{V}$ is *expressive and diverse*, containing roughly $50,000$ tokens, ranging from famous personalities to emojis and even names of video games (*e.g.*, Fortnite is a single token in $\mathcal{V}$). By learning our decomposition over this rich set of candidate concepts, we facilitate a meaningful yet non-restrictive optimization over semantic, human-understandable information. Thus, our choice of candidate concepts achieves the best of both worlds – it is both expressive and interpretable. Second, we make a design choice to *optimize the coefficients as a function of the word embeddings*. This choice is critical to the success of our method since it utilizes the rich textual embedding space of CLIP in determining the coefficients. Effectively, this reduces the optimization problem from optimizing $50,000$ unrelated coefficients to learning a mapping from a smaller semantic space, $\mathbb{R}^d$. Specifically, our method assigns a coefficient $\alpha$ for each word embedding $w$ using a learned 2-layer MLP, which takes as input the word embedding vector $w$ as follows,

$$\forall w \in \mathcal{V}: \ \alpha = f(w) = \sigma\left(W_2(\sigma(W_1(w)))\right), \tag{4}$$

where $\sigma$ is the ReLU non-linearity (Agarap, 2018), and $W_1, W_2$ are linear mappings. Based on $f$, we compute $w_N^* = \sum_{i=1}^{N} f(w_i)w_i$. Note that this pseudo-token is not identical to the output token $w^*$ since $w_N^*$ contains all tokens in $\mathcal{V}$. $w^*$ is obtained by the top tokens from $w_N^*$, as described in Eq. 5.

To learn a meaningful pseudo-token $w^*$, we optimize our MLP to reconstruct the images generated from $\mathcal{P}^c$. This choice encourages our pseudo-token to imitate the denoising process of the concept images. We draw a random noise $\varepsilon \sim \mathcal{N}(0,1)$ and a timestep $t \in \{1, \ldots, T\}$ for each image, and noise the images according to Eq. 2. We then employ the model's reconstruction objective from Eq. 1.

As mentioned, $w_N^*$ considers all the tokens in the vocabulary. However, for better interpretability, we wish to represent the input concept with a *small* set of $n << N$ tokens. Notate by $w_1, \ldots, w_n \in \mathcal{V}$ the tokens with the highest learned coefficients. We add a regularization loss to encourage the pseudo-token $w_N^*$ to be dominated by these top $n$ tokens, *i.e.*,

$$\mathcal{L}_{sparsity} = 1 - cosine\left(w^*, w_N^*\right). \tag{5}$$

This encourages the pseudo-token $w^*$, defined by the top $n$ tokens in $\mathcal{V}$, to be semantically similar to $w_N^*$, which is defined by the entire vocabulary. Our overall objective function is, therefore,

$$\mathcal{L} = \mathcal{L}_{reconstruction} + \lambda_{sparsity}\mathcal{L}_{sparsity}, \tag{6}$$

In our experiments, we set $\lambda_{sparsity} = 0.001, n = 50$. At inference time, we employ the MLP on the vocabulary $\mathcal{V}$ and consider the top $n = 50$ tokens to compose $w^*$, as specified in Eq. 3. Implementation details and a figure describing our method can be found in Appendix A.

**Single-image decomposition** Given an image $I$ that was generated by SD for a concept $c$, we wish to determine the subset of the tokens from the decomposition $w^*$, that drove the generation of this specific image. This is done via an iterative process over the tokens $w_j \in w^*$ as follows; at each step, we attempt to remove a single token from the decomposition, $w_j^* = \sum_{i \neq j} \alpha_i w_i$, and generate the corresponding image $I_j$ with the prompt $\mathcal{P}^{w_j^*}$ and the same seed. Next, we use CLIP's image encoder to determine if $I_j$ is semantically identical to $I$. If the CLIP score of the two images is higher than 95, we remove $w_j$ from $w^*$ and continue to the next token. This criterion avoids incorporating tokens whose removal only causes minor non-semantic modifications to the image $I$ (such as a slight shift in pose). This process is repeated for all tokens in $w^*$ until no tokens are removed.

## 4 EXPERIMENTS

**Concept-based explanation desiderata** Following previous literature on concept-based explainability (Ghorbani et al., 2019), we begin by defining a set of desired properties that concept-based explanations for diffusion models should uphold. These properties will be the criteria to evaluate our method. (i) **Meaningfulness**- Each decomposition element should be semantically meaningful and human-understandable. (ii) **Faithfulness**- The decomposition should be faithful to the concept

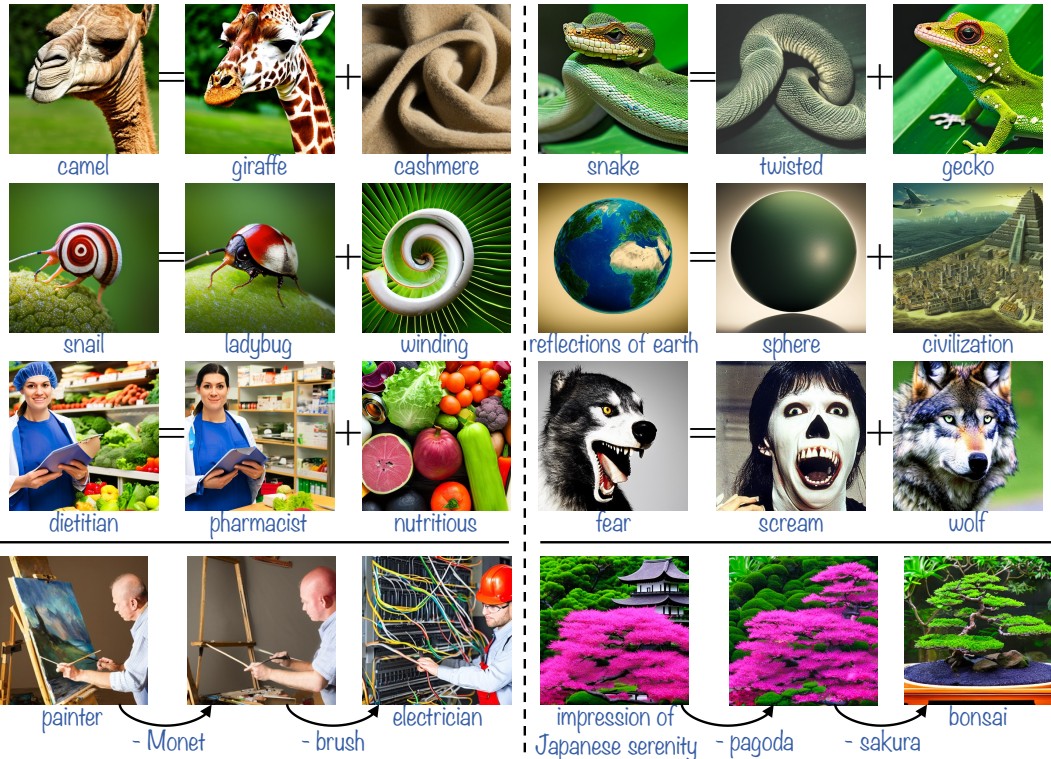

Figure 2: Decompositions of single images by CONCEPTOR. The top rows present images found to contain two elements. The last row shows more complex mixtures by removing one element at a time. The examples demonstrate the *meaningfulness* of our learned decompositions.

representation by the model. In other words, the decomposition should reconstruct the features manifested in each of the concept images, and produce images that are in the distribution of the concept images. (iii) **Robustness**- The decomposition should be independent of the selection of the training set and the initialization. Next, we conduct extensive experiments to demonstrate our method's ability to produce meaningful, faithful, and robust decompositions for diverse types of concepts. Throughout this section, we notate by $w^c$ the token(s) corresponding to the concept $c$.

**Data** We construct a diverse and comprehensive dataset of 188 concepts, comprised of the basic classes from CIFAR-10 (Krizhevsky, 2009), a list of 28 professions from the Bias in Bios dataset (De-Arteaga et al., 2019), 10 basic emotions and 10 basic actions, all 30 prompts from the website *Best 30 Stable Diffusion Prompts for Great Images*[*], which contains complex prompts that require hierarchical reasoning (*e.g.*, *"Medieval village life"*, *"impression of Japanese serenity"*), and, finally, we consider 100 random concepts from the ConceptNet (Speer & Havasi, 2013) knowledge graph to allow for large-scale evaluation of the methods. A full list of all concepts is provided in Appendix B.

### 4.1 QUALITATIVE RESULTS

The ability to link between textual decomposition elements and their visual manifestation is important to facilitate human understanding and establish *meaningfulness*. We propose two strategies to obtain these connections. First is our single-image decomposition scheme, described in Sec. 3. Second, we propose to gradually manipulate the element's coefficient to observe the visual changes it induces.

**Single-image decomposition** Figs. 1 and 2 and Appendix D contain examples of decompositions over images generated by SD. The first three rows of Fig. 2 present examples of images that decompose into two elements, while the last row contains more complex concepts that decompose into three elements, where we visualize the removal of one element at a time. Note that the results show non-trivial and profound links between concepts. For example, the *"snake"* in Fig. 2 is constructed as a *"twisted gecko"*, the *"camel"* borrows its skin texture and color from the *"cashmere"*, *etc*. These

---

[*]https://mspoweruser.com/best-stable-diffusion-prompts/

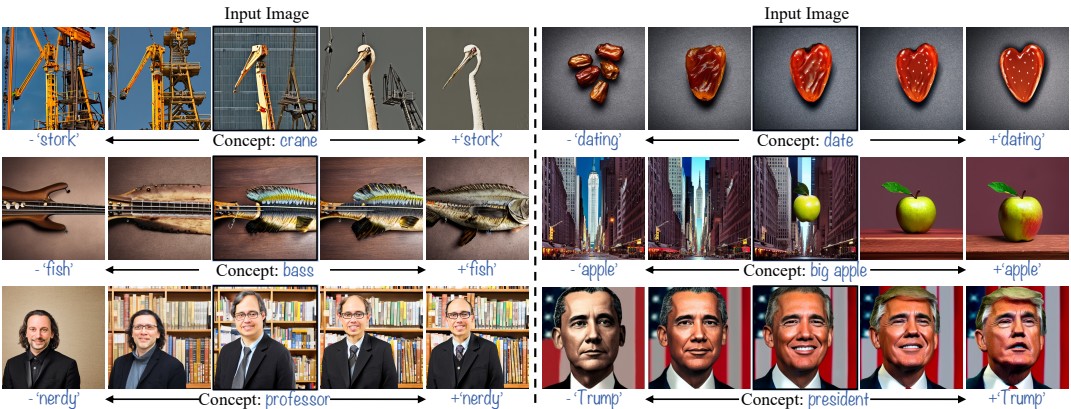

Figure 3: Coefficient manipulation. For each of the listed concepts, we manipulate the coefficient of a single element from the decomposition and observe its visual impact on the generated image.

Table 1: Quantitative evaluation of CONCEPTOR and the baselines.

| Method | CLIP pairwise↑ | LPIPS↓ | FID per concept↓ | FID entire set↓ | Token diversity |
|---|---|---|---|---|---|
| BLIP-2 token | $66.3 \pm 16.8$ | $0.60 \pm 0.13$ | $218.1 \pm 93.4$ | 46.6 | $52.1 \pm 9.7$ |
| BLIP-2 sentence | $78.7 \pm 11.2$ | $0.57 \pm 0.8$ | $158.6 \pm 74.0$ | 23.7 | $65.8 \pm 1.9$ |
| PEZ | $79.1 \pm 9.9$ | $0.56 \pm 0.06$ | $150.6 \pm 63.4$ | 18.2 | $\mathbf{75.9} \pm 1.2$ |
| NMF | $80.0 \pm 9.7$ | $0.53 \pm 0.06$ | $147.0 \pm 68.9$ | 21.6 | — |
| k-means | $82.5 \pm 8.4$ | $0.53 \pm 0.05$ | $132.5 \pm 60.0$ | 21.3 | — |
| PCA | $83.0 \pm 8.0$ | $0.53 \pm 0.09$ | $130.8 \pm 52.9$ | 19.8 | — |
| **CONCEPTOR** | $\mathbf{86.2} \pm 8.3$ | $\mathbf{0.44} \pm 0.09$ | $\mathbf{109.5} \pm 51.8$ | $\mathbf{9.8}$ | $69.8 \pm 3.4$ |

examples demonstrate associations beyond textual correlations, based on visual similarities such as shape, texture, and color. Additionally, note that the decompositions represent *various different representation strategies*. Some involve a mixture of shape and appearance (*e.g.*, *"snail"*), others involve compositional features added gradually (*e.g.*, *"impression of Japanese serenity"*), *etc*. Finally, observe that the *"painter"* decomposition demonstrates reliance on renowned artistic styles, such that when *"Monet"* is removed, the painting disappears, even though *"Monet"* did not appear explicitly in the input prompt. Appendix D.1 presents a further investigation of this reliance on artistic styles.

**Coefficient manipulation**    Fig. 3 presents examples of coefficient manipulations. First, we present examples of homograph concepts (first, second row of Fig. 3). Observe that in some cases, *e.g.*, *"big apple"*, both meanings are generated separately in the image (an apple, New York City), while other cases, *e.g.*, *"crane"*, *"date"* generate a single object. Even in the latter case, the manipulation shows that *both meanings impact the generated image, implicitly*. For example, when reducing the element *"stork"* from *"crane"*, the structure of the crane changes. Evidently, the model employs both meanings simultaneously, borrowing the appearance from the machine and the shape from the bird.

Next, we present examples of element inspection. In the first example, we visualize the impact of *"nerdy"* on the concept *"professor"*. As can be observed, it controls the professor's baldness, the glasses, the suit, and the library in the background. Secondly, we inspect exemplar interpolations. Observe that the element *"Trump"* in an image of a *"president"* controls semantic features borrowed from the identity of Donald Trump while removing this element results in a president that resembles Obama. This example directly demonstrates an interpolation of exemplars. Features from both Obama and Trump are employed *simultaneously* in the image generation process. This phenomenon suggests that diffusion models can also memorize by mixing inputs, beyond exact single-sample reconstruction. Further investigation of exemplar-based representations can be found in Appendix E.

## 4.2    QUANTITATIVE RESULTS

**Baselines**    As far as we can ascertain, our work is the first to tackle concept representations in text-to-image diffusion models. We, therefore, compare our method with reasonable baselines and adaptations of existing methods. First, we consider a prompt tuning method, *Hard Prompts Made Easy (PEZ) (Wen et al., 2023)*, which aims to learn a prompt that will reproduce an input

Table 2: Qualitative comparison of the decompositions by our method and the leading baselines.

| Concept | PEZ | BLIP-2 sentence | CONCEPTOR |
|---|---|---|---|
| Rapper | `marin, prodigy, sturridge, noneeminem` | man, hat, shirt | `Tupac, Drake, Khalifa, Weekend, Khaled, hood, hat` |
| Medieval village life | `moderated, humpday, giftideas, shistory` | people, medieval clothing, village | `caravan, peasant, medieval, countryside, farming` |
| Elegance on a plate | `silver, chesterfield, dinner, moschbizitalk` | plate, vegetable, meat | `plating, chef, beauty, dessert, saucer, porcelain` |

set of training images. Second, we consider two baselines that leverage the state-of-the-art image captioning model BLIP-2 (Li et al., 2023a): (i) *BLIP-2 sentence* extracts a single caption per concept by decoding the mean CLIP embedding of the training images. (ii) *BLIP-2 token* creates one caption per image and constructs a single pseudo-token from the captions, where each token is weighted by its frequency in the captions. Finally, we consider CNN-based concept interpretability methods. As mentioned in Sec. 2, these methods are not directly applicable to diffusion models. Therefore, we present comparisons to the closest adaptations of these methods. We decompose the activation of the UNet features from different denoising steps into concept activation vectors (CAVs) using *k-means (Ghorbani et al., 2019), PCA (Zhang et al., 2020)* and *NMF (Fel et al., 2022)*. At inference, we project the intermediate activations into the learned space, see Appendix A.1 for more details.

**Metrics** For each concept, we test the *faithfulness* and the *diversity* of the decompositions. We use a test set of 100 seeds to generate images with $w^c$ and with each method. Then, we employ three types of metrics: (i) *Pairwise Similarity*, to measure the faithfulness of the decomposition w.r.t. each of the concept images. We report the mean CLIP (Radford et al., 2021) image similarity and the LPIPS (Zhang et al., 2018) score. (ii) *Distribution similarity*, to measure the faithfulness of the decomposition w.r.t. the concept distribution. We report the FID (Heusel et al., 2017) score with respect to the concept images for each concept separately (*FID per concept*) and for all concepts as a dataset (*FID entire set*). (iii) We employ SentenceBERT (Reimers & Gurevych, 2019) to measure the element diversity by estimating the dissimilarity of the tokens in the decomposition (*Token diversity*). This metric further substantiates the *meaningfulness* by showing that the decomposition is diverse.

**Results** The results, averaged across all 188 concepts, are reported in Tab. 1. As can be seen, our method outperforms all baselines across all faithfulness metrics. Notably, our LPIPS score is at least 10% lower than that of all baselines, indicating that our method is faithful to the concept images. Importantly, both FID metrics obtained by our method are the lowest by a big margin (at least 20 per concept and 8 for the entire set). The scales of the FID scores are very different since the per concept FID was calculated on sets of merely 100 images. However, we report both metrics for completeness. Focusing on the more reliable FID score on the entire set, we calculate the "ground-truth" FID between the train and test set and obtain a score of 7.1, which is fairly close to our score. These results establish CONCEPTOR's ability to provide *faithful* decompositions. Considering the scores by SentenceBERT, CONCEPTOR is only superseded by PEZ. However, PEZ produces a significant amount of uninterpretable tokens (see Tab. 2). The matrix factorization methods do not produce text, therefore we cannot compute the SentenceBERT score. Instead, we enclose in Appendix I the top principal components learned for the concepts from Fig. 4. As can be seen, the obtained components do not appear to be coherent or interpretable, *i.e.*, these methods violate the *meaningfulness* criterion.

Next, we conduct qualitative comparisons between CONCEPTOR and the leading baselines. Tab. 2 compares the textual decompositions, showing that CONCEPTOR learns diverse and meaningful decompositions. Some concepts, such as *"a rapper"* are dominated by exemplars (*e.g. "Drake"*, *"Tupac"*), while others, such as *"Medival village life"*, are a composition of semantically related concepts (*e.g.*, *"caravan"*, *"peasant"*). In contrast, the baselines either produce decompositions that are not interpretable, *i.e.*, violate the *meaningfulness* criterion (PEZ), or are oversimplistic (BLIP-2). Please refer to Appendix F for word cloud visualizations of CONCEPTOR over complex prompts. Fig. 4 presents a visual comparison to the leading baselines given the same seeds. As can be observed, CONCEPTOR successfully preserves the image features (*i.e.*, upholds *faithfulness*), even when the concept entails detailed features. For example, the *"painter"* images demonstrate a *reconstruction of the paintings*. Conversely, the baseline methods do not accurately embody all features of the concept.

**User study** To further substantiate our *meaningfulness* and *humans' ability* to understand our decompositions, we conduct a user study. In the study, we randomly draw 8 concepts, 2 from each of our data categories: (a) professions, (b) abstract concepts, (c) basic concepts (ConceptNet,

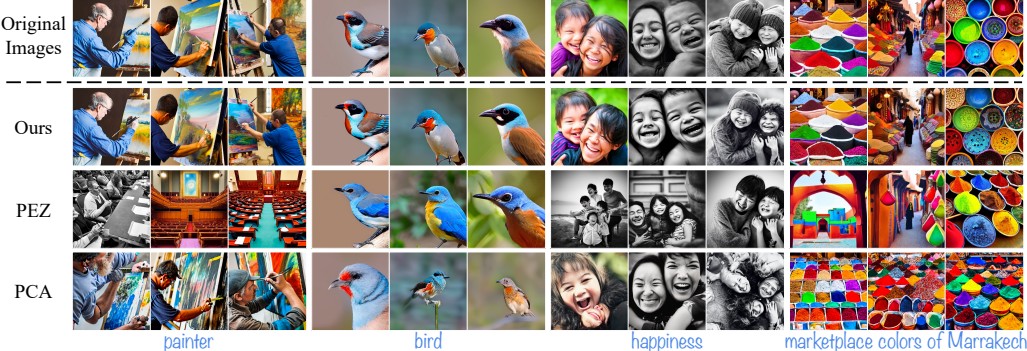

Figure 4: Feature reconstruction comparison to the leading baselines. For each concept (column) we generate the images using the same random noise with our method and the leading baselines, and compare to the original concept images generated by Stable Diffusion (Original Images).

Table 3: Ablation study of our method, conducted on the professions subset (De-Arteaga et al., 2019).

| Method | CLIP pairwise↑ | LPIPS↓ | FID per concept↓ | Token diversity↑ |
|---|---|---|---|---|
| CONCEPTOR | **87.0** ± 5.5 | **0.45** ± 0.07 | **107.96** ± 31.0 | 69.7 ± 3.4 |
| w/o MLP | 78.0 ± 6.7 | 0.55 ± 0.06 | 142.88 ± 45.1 | **75.9** ± 3.0 |
| w/o Eq. (5) | 80.3 ± 11.6 | 0.52 ± 0.09 | 146.4 ± 63.4 | 73.2 ± 2.1 |
| $n = 10$ | 82.9 ± 7.8 | 0.49 ± 0.11 | 129.41 ± 55.3 | 54.6 ± 9.4 |
| $n = 100$ | 85.6 ± 6.9 | 0.47 ± 0.07 | 114.36 ± 39.7 | 72.8 ± 1.8 |
| CLIP top words | 80.1 ± 9.9 | 0.513 ± 0.1 | 130.9 ± 57.2 | 66.3 ± 3.9 |

CIFAR-10), and (d) complex concepts. For each concept, the users were presented with 6 random concept images and 3 possible textual decompositions, as provided by CONCEPTOR, and the two leading baselines that extract text (*BLIP-2 sentence* and *PEZ*). The users were asked to select the decomposition that best captures all the features presented in the images. Overall, we collected 160 responses for all questions. Of those 160 responses, 70% favored the decomposition by CONCEPTOR above all alternatives, 26.9% favored the captioning by BLIP-2, and 3.1% selected PEZ, further establishing CONCEPTOR's meaningfulness, even compared to the natural captioning alternative.

**Robustness experiments**     In Appendix C we conduct extensive *robustness* tests, to demonstrate two important properties: (i) CONCEPTOR is robust to *different choices of the training set and initialization*, and (ii) CONCEPTOR generalizes the training task to *test images*. Experiment (i) employs 3 different training sets and initializations and tests the *exact* match of tokens between the 3 resulting decompositions. The results demonstrate that the decomposition is consistent across all sets- $72 - 80\%$ of the top-10 elements, and $63 - 70\%$ of the top-25 elements are preserved across all choices. Experiment (ii) shows that $w^*$ is able to denoise *test* concept images from *any denoising step*, thus $w^*$ indeed captures the concept features beyond the selection of training images.

In conclusion, we find that the baselines either violate the *meaningfulness* criterion, *i.e.*, produce uninterpretable decompositions (*k-means, PCA, NMF, PEZ*, see Tab. 2 and Appendix I, user study) or the *faithfulness* criterion (*BLIP-2*, see poor performance in Tab. 1). We note that for both criteria and in all experiments, CONCEPTOR significantly outperforms all baselines and obtains *robust*, *faithful*, and *meaningful* decompositions (see Tabs. 1 and 2, Fig. 4, and Appendix C, user study).

**Ablation Study**     We conduct an ablation study to examine the impact of each component on our method. First, we ablate the choice of employing an MLP to learn the coefficients and instead learn them directly. Next, we ablate our sparsity loss and the choice of $n = 50$. Last, we ablate our choice of vocabulary $\mathcal{V}$ and instead extract the top $50$ tokens by their CLIP similarity to the mean image.

The results are listed in Tab. 3. Replacing the MLP with a vector of weights is *detrimental to all faithfulness metrics*. This demonstrates the importance of our learned MLP, as it leverages the rich semantic information learned by CLIP, rather than optimizing a huge set of coefficients without any semantic understanding. Without the sparsity loss (Eq. (5)), the top $50$ tokens do not necessarily reflect the learned token $w_N^*$, and all metrics except for token diversity deteriorate. Additionally, observe that the performance decreases when employing $n = 10$ since the decomposition is not rich enough to represent all features. For $n = 100$, the results are similar to the full method, other than the diversity which improves a little. This indicates that CONCEPTOR is relatively stable to this parameter.

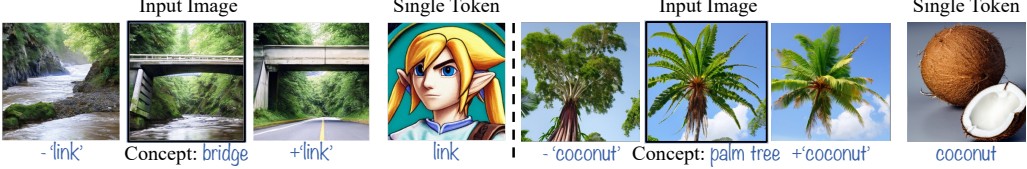

Figure 5: Examples of elements that impact the generation differently from their standalone meaning.

Finally, when only considering the top words by CLIP similarity to the images, the performance decreases substantially, supporting the reliance of our method on a wide variety of tokens from the vocabulary, beyond the ones most correlated with the images in terms of textual semantics.

### 4.3 BIAS DISCOVERY AND MITIGATION

Another important capability of our method is bias discovery. Text-to-image models have demonstrated the representation of social biases (Luccioni et al., 2023). CONCEPTOR can be used to discover such biases through the decomposition elements. Tab. 4 lists examples of such biased concepts. Note that our method detects behaviors that are not necessarily observable visually such as a connection between *"Jews"* and *"journalists"*. These findings

Table 4: Biases revealed by CONCEPTOR.

| Concept | Examples of biased terms |
|---|---|
| Secretary | `womens, girl, ladies` |
| Opera singer | `obese, overweight, fat` |
| Pastor | `Nigerian, gospel` |
| Journalist | `refugee, jews` |
| Drinking | `millennials, blonde` |

underscore the importance of researching concept representations in text-to-image models, as biases can impact the generation, even if visually elusive. Utilizing our method, one can also choose to generate debiased versions of these concepts by decreasing coefficients associated with biased tokens *while preserving other features in the scene*. Please refer to Appendix H for such examples.

## 5 DISCUSSION AND LIMITATIONS

While our method provides faithful and interpretable concept decompositions, there are some limitations to consider. First, we find that the visual impact of an element is not always completely aligned with its impact as a single token, *i.e.*, *the impact of each token depends on the context of the decomposition*. Fig. 5 demonstrates such cases. For each concept, we visualize the effect of manipulating the token, and the result of keeping only the token of interest (Single Token). Note that the influence of a token on the generated image differs from its influence as a sole token. For example, the token *"link"* on its own produces an image of the video game character, Link. However, in the context of a *"bridge"*, it adds a solid bridge (a *link*) between the two edges of the image.

Second, our method is limited to elements that are single tokens, therefore a complex phrase (*e.g.*, *"palm tree"*) will not be included as a single element in our decomposition. However, as mentioned, our construction of the linear combination mitigates the impact of this limitation. Since the context of the decomposition can change the meaning of an element, complex relations can be formed between single tokens by leveraging the context. This is exemplified by our ability to decompose complex concepts, that require hierarchical reasoning (*e.g.*, *"elegance on a plate"*, *"rainbow dewdrops"*).

## 6 CONCLUSIONS

How does a generative model perceive the world? Focusing on text-to-image diffusion models, we investigate the model's internal knowledge of real-world concepts. We present CONCEPTOR, a method to provide a human-understandable decomposition for a textual concept. Through extensive experiments, we show that CONCEPTOR provides interpretations that are meaningful, robust, and faithful to the concept representation by the model. Using CONCEPTOR, we obtain various interesting observations on the learned concept representations. Via a per-image decomposition scheme, we observe non-trivial connections between concepts in ways that transcend the lexical meaning of the tokens. Furthermore, our method exposes less intuitive behaviors such as the reliance on exemplars, mixing dual meanings of concepts, or non-trivial biases. In all cases, the novel paradigm allows us to shed new light on a model that, similar to other foundation models, can still be considered an enigma.

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

# A    IMPLEMENTATION DETAILS

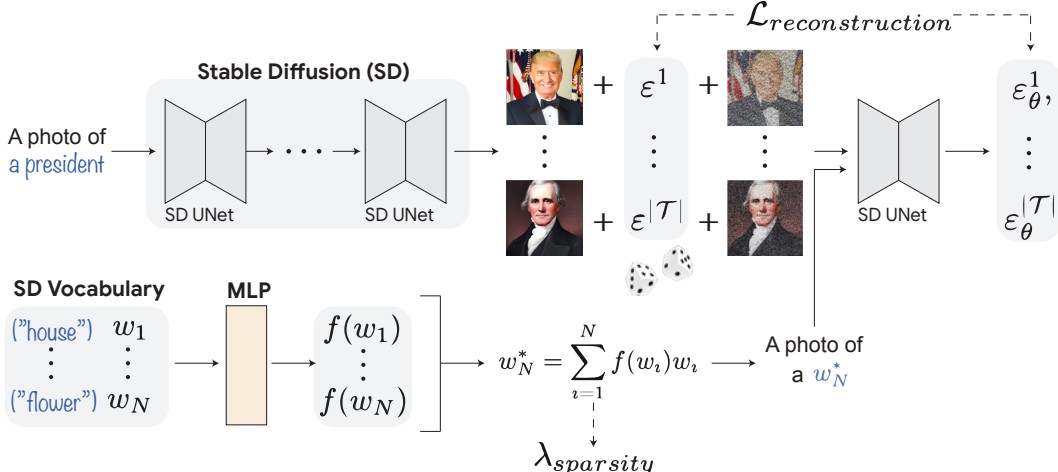

Figure 6: Illustration of the CONCEPTOR method. Given the concept of interest (*e.g.*, *"a president"*), we generate 100 concept images. Next, a learned MLP network maps each word embedding $w_i$ to a coefficient $f(w_i)$, and the pseudo token $w_N^*$ is constructed as a linear combination of the vocabulary. We then add random noises $\varepsilon^1, \ldots, \varepsilon^{|\mathcal{T}|}$ to the images, and use the model to predict the noise based on the text *"a photo of a <$w_N^*$>"*. We train the MLP with the objective of reconstructing the images ($\mathcal{L}_{reconstruction}$) and add a sparsity loss to encourage sparse coefficients ($\mathcal{L}_{sparsity}$).

Fig. 6 describes the pipeline of our method. Given an input concept (*e.g.*, *"a president"*), we obtain a set of representative concept images. At each optimization iteration, we add random noises to each of the images individually. Our learned MLP operates over the model's vocabulary to produce a coefficient for each of the word embeddings, and the resulting pseudo-token is used to denoise the training images. We employ a reconstruction loss to encourage the pseudo-token to contain the concept elements learned by the model and add a sparsity loss to maintain an interpretable and stable decomposition.

All of our experiments were conducted using a single A100 GPU with 40GB of memory. We train our MLP as specified in Sec. 3 of the main paper with 100 images generated from the concept using seed 1024 for a maximum of 500 training steps with a batch size of 6 (which is the largest batch size that could fit on our GPU). Additionally, we use a learning rate of $1e-3$ (grid searched on 5 concepts between $1e-2, 1e-3, 1e-4$). We conduct validation every 50 optimization steps on 20 images with a validation seed and select the iteration with the best CLIP pairwise similarity between the reconstruction and the concept images. We use the latest Stable Diffusion v2.1[*] text-to-image model employing the pre-trained text encoder from the OpenCLIP ViT-H model[*], with a fixed guidance scale of 7.5.

Additionally, to filter out meaningless tokens such as punctuation marks we consider the vocabulary to be the top $5,000$ tokens by their CLIP similarity to the mean training image. We note that this filtering method is fairly coarse, and meaningless tokens remain in the vocabulary, however, we find that this choice improves the convergence time of our MLP such that 500 iterations are enough to obtain meaningful decompositions. This choice is ablated in the main paper (see the CLIP top words ablation). Our single-image decomposition scheme employs a CLIP ViT-B/32 model (Radford et al., 2021). Please find our code attached as a ZIP file to reproduce our results.

## A.1    CONCEPT-BASED BASELINES IMPLEMENTATION DETAILS

To implement the concept-based baseline approaches, we follow the formulation in Fel et al. (2023). During the generation of the training concept images, we collect the intermediate representations from

---

[*]https://github.com/Stability-AI/stablediffusion
[*]https://github.com/mlfoundations/open_clip

the UNet at different denoising timesteps. Then, we decompose those representations using PCA, following Zhang et al. (2020), NMF, following Fel et al. (2022) or k-means, following Ghorbani et al. (2019). We use a total of $n = 50$ components, to match our decomposition size. Finally, at inference, we generate images from the concept using the test seed, 2, while projecting and reconstructing the aforementioned intermediate representations both before passing them to the next block and to the upsampling (decoding) part of the UNet (via the skip-connections). At training and inference when using NMF we drop the negative activations. At inference, when using k-means, we employ the closest cluster as a reconstruction of an intermediate representation.

- We compute PCA using 5 iterations of randomized SVD solver with 10 oversamples.
- We compute k-means using 10 k-means++ initialization and 100 iterations of the Lloyd algorithm.
- We compute NMF using NNDSVD initialization and run the Coordinate Descent solver for 200 iterations.

## B  DATASET

In the following, we enclose the full list of concepts in each of our data subsets.

**Bias in Bios concepts**   *"professor"*, *"physician"*, *"attorney"*, *"photographer"*, *"journalist"*, *"nurse"*, *"psychologist"*, *"teacher"*, *"dentist"*, *"surgeon"*, *"architect"*, *"painter"*, *"model"*, *"poet"*, *"filmmaker"*, *"software engineer"*, *"accountant"*, *"composer"*, *"dietitian"*, *"comedian"*, *"chiropractor"*, *"pastor"*, *"paralegal"*, *"yoga teacher"*, *"dj"*, *"interior designer"*, *"personal trainer"*, *"rapper"*

**CIFAR-10 concepts**   *"airplane"*, *"automobile"*, *"bird"*, *"cat"*, *"deer"*, *"dog"*, *"frog"*, *"horse"*, *"ship"*, *"truck"*

**Emotions**   *"affection"*, *"anger"*, *"disgust"*, *"fear"*, *"happiness"*, *"honor"*,*"joy"*, *"justice"*, *"sadness"*, *"beauty"*

**Actions**   *"clapping"*, *"climbing"*, *"drinking"*, *"hugging"*,*"jumping"*, *"pouring"*, *"running"*, *"sitting"*, *"throwing"*, *"walking"*

**Complex concepts**   *"Rainy New York Nights"*, *"The Fashion of Abandoned Places"*, *"Rainbow Dewdrops"*, *"Aerial Autumn River"*, *"Skateboarder's Urban Flight"*, *"Dive into Coral Reefs"*, *"Vintage European Transit"*, *"Star Trails over Mountains"*, *"Marketplace Colors of Marrakesh"*, *"Elegance on a Plate"*, *"The Renaissance Astronaut"*, *"The Surreal Floating Island"*, *"Impression of Japanese Serenity"*, *"Jazz in Abstract Colors"*, *"The Confluence of Pop Art"*, *"The Robotic Baroque Battle"*, *"Cubist Bustling Market"*, *"The Romantic Stormy Voyage"*, *"The Botanist in Art Nouveau"*, *"The Gothic Moonlit Castle"*, *"Neon-Soaked Cyberpunk City"*, *"Dragon's Stormy Perch"*, *"Reflections of Earth"*, *"After The Fall"*, *"Retro Gaming Nostalgia"*, *"Medieval Village Life"*, *"Samurai and the Mystical"*, *"Minimalistic Geometry"*, *"Alien Flora and Fauna"*, *"The Inventor's Steampunk Workshop"*

**ConceptNet concepts**   *"curling iron"*, *"baseball stadium"*, *"flowers"*, *"submarine"*, *"policeman"*, *"projectile"*, *"tissue holder"*, *"jogging"*, *"storey"*, *"sickness"*, *"parlor"*, *"ships"*, *"conductor"*, *"booze"*, *"key"*, *"metal"*, *"prostitute"*, *"wings"*, *"tools"*, *"road"*, *"main"*, *"leader"*, *"radio"*, *"surprise"*, *"chips"*, *"castle"*, *"bathroom"*, *"compete against"*, *"leather"*, *"science"*, *"rich"*, *"sponge"*, *"bell"*, *"eloquent"*, *"nightclub"*, *"water"*, *"patient"*, *"eat vegetables"*, *"respect"*, *"lemur"*, *"bum"*, *"mammoth"*, *"birthday"*, *"chain"*, *"cats"*, *"frogs"*, *"arkansas"*, *"basketball"*, *"listening"*, *"dream"*, *"ticket office"*, *"failure"*, *"text"*, *"now"*, *"oven"*, *"leg"*, *"mundane"*, *"copulate"*, *"tree"*, *"wood"*, *"mail"*, *"wooden rod"*, *"clippers"*, *"competing against"*, *"dull"*, *"book"*, *"watch television"*, *"winning baseball game"*, *"iphone"*, *"dance club"*, *"security"*, *"politician"*, *"subway station"*, *"fall"*, *"junk"*, *"sleighing ride"*, *"call"*, *"mosquitoes"*, *"national highway"*, *"contraceptive device"*, *"statement"*, *"kill"*, *"seeing old things"*, *"lift"*, *"adults"*, *"pillowcase"*, *"wedding ring"*, *"eyes"*, *"country"*, *"stepladder"*, *"mandolin"*, *"reception area"*, *"chief"*, *"plastic"*, *"projector"*, *"hub"*, *"card catalog"*, *"negligible"*, *"rook"*, *"llano estacado"*

# C ROBUSTNESS EXPERIMENTS

In the following sections, we conduct experiments to demonstrate our method's ability to provide robust concept interpretations. First, we show that the obtained decomposition is stable, *i.e.*, that the same elements are learned across different choices of training sets and initialization. Second, we test our decomposition's ability to generalize w.r.t. the denoising training task, *i.e.*, we test the denoising capabilities of $w^*$ on test images. This shows that the elements in $w^*$ represent the entire concept, beyond the training images.

## C.1 ROBUSTNESS TO TRAINING DATA

Note that since $N >> d$, there are many linear combinations that yield $w^*$. However, due to the specific MLP-based structure and the sparsity constraints, the decomposition is stable across multiple draws of training sets. In this section, we aim to verify that empirically. For each concept, we generate 2 alternative training sets with different random seeds, in addition to our original training set, to test the consistency of our results. For each alternative training set, we decompose the concept using our method as described in Sec. 3, with a different initialization for the MLP*. This process results in 3 decompositions of $n = 50$ tokens for each concept.

We then analyze the intersection of the top $k = 10, 25$, and 50 tokens between the original decomposition and each of the alternative decompositions. The concept intersection score for $k$ is defined to be the average of the intersections with the two alternative sets. In other words, we calculate two intersection sizes for $k$: between the top $k$ tokens of the original decomposition and the first alternative decomposition, and between the top $k$ tokens of the original decomposition and the second alternative. The overall concept intersection score for $k$ is the average of the two. Standard Deviation is computed across the concepts. Note that this experiment measures an *exact match* of the tokens, therefore the actual intersection may be even higher, *e.g.*, if a synonym is used in one of the alternatives.

Table 5: Decomposition consistency experiment. For each number of tokens ($k = 10, 25, 50$) we test the intersection between our learned top tokens and those learned by employing two *different* training sets of concept images, with different random initializations. The results demonstrate that the top tokens are consistent and robust across different training sets and seeds.

|  | No. of Tokens | Intersection |
|---|---|---|
| **Concrete** | Top 10 | $8.03 \, (80.3\%) \pm 2.07$ |
|  | Top 25 | $17.68 \, (70.7\%) \pm 4.47$ |
|  | Top 50 | $28.96 \, (57.9\%) \pm 8.05$ |
| **Abstract** | Top 10 | $7.20 \, (72.0\%) \pm 1.86$ |
|  | Top 25 | $15.95 \, (63.8\%) \pm 3.97$ |
|  | Top 50 | $25.65 \, (51.3\%) \pm 5.41$ |

The average intersection scores across all concrete concepts and all abstract concepts are presented in Tab. 5. As can be seen, for the concrete concepts, an average of $8.03(80.3\%)$ of the top 10 tokens are present in all the decompositions, even when considering an entirely different training set, indicating that the top tokens obtained by our method are stable. Additionally, when considering the top 25 tokens, an average of $17.68(70.7\%)$ of the tokens are present in all decompositions, which is a large majority. We note that the bottom tokens are typically less influential on the decomposition, as they are assigned relatively low coefficients by the MLP. Accordingly, when considering all 50 tokens, an average of $28.96(57.9\%)$ of the tokens appears in all decompositions. The results for the abstract concepts are slightly lower, yet demonstrate a similar behavior. Overall, the results demonstrate that our method is relatively robust to different training sets and random seeds, such that even in the face of such changes, the top-ranked tokens remain in the decomposition.

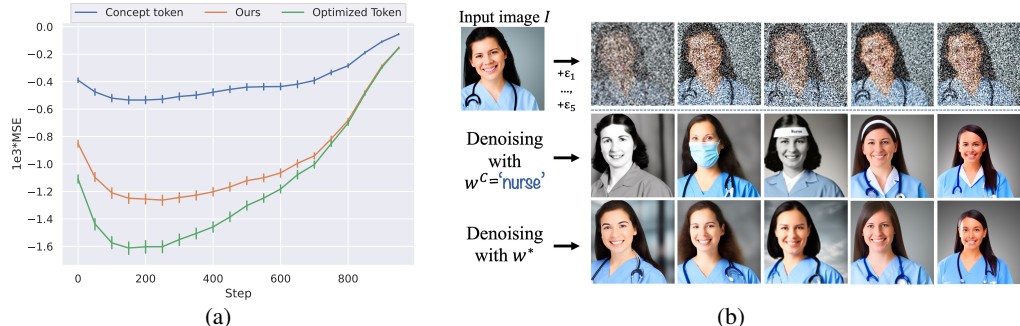

(a)                                                                 (b)

Figure 7: Generalization tests comparing the concept prompt, $w^c$, and our pseudo-token, $w^*$. (a) Quantitative test on all concepts. For each timestep, we add random noises to the images and compare the reconstruction with $w^*$, $w^c$, and $w^o$, a continuous token optimized with the same objective as $w^*$ (Optimized Token). We report the MSE after subtracting the score of a random token. (b) Qualitative comparison. An image $I$ is generated from the concept *"a nurse"*, and different noises are added $\varepsilon_1, \ldots, \varepsilon_5$ (1st row). We then compare the denoising with $w^c$ (2nd row) and with $w^*$ (3rd row).

## C.2    DENOISING GENERALIZATION

We wish to ascertain that the learned decomposition $w^*$ can generalize its training task to test images. Recall that during training, $w^*$ accumulated the elements necessary to denoise the concept images *for each denoising step*. Next, we wish to demonstrate that these features can perform this same training task on test images. We compare the denoising quality with $w^*$ to that of the concept prompt token(s), $w^c$. This comparison serves two purposes: (1) establish CONCEPTOR's ability to learn the actual concept features, beyond a set of features that simply reconstruct the training set, (2) motivate the fact that, as can be seen in our results, $w^* \neq w^c$. To intuitively motivate the latter, note that even though $w^c$ was used to generate the concept images, it is not necessarily the best prompt to denoise them, since: (1) $w^c$ generates each image using a specific initial random noise, but is not guaranteed to be better in denoising them after applying other random noises. (2) Unlike $w^c$, $w^*$ is constructed as a *linear combination of tokens*. Thus, our optimization is performed over a larger, *continuous* space of embeddings, and therefore is more expressive than a simple selection of tokens.

We begin with a quantitative comparison. We sample a test set of 100 images for each concept in the dataset. Then, for each denoising step $t \in \{1, \ldots, T\}$ and each test image, we draw a random noise and apply it as in Eq. 2. Finally, we test the reconstruction loss specified in Eq. 1 with the pseudo-token $w^*$ compared to the concept prompt $w^c$. To provide a lower bound on the obtainable error, we additionally compare to $w^o$, a vector optimized with the same reconstruction objective on the entire continuous embedding space $\mathbb{R}^d$ without restrictions, similar to (Gal et al., 2022). Note that, unlike $w^*$, $w^o$ does not offer interpretable information, since it is non-decomposable, and often out-of-distribution (see Sec. 2). However, it demonstrates our claim that optimization over a larger domain yields better reconstruction. Additionally, we observe that there is a large variance in the MSE score across timesteps. Latents in early steps are very noisy, and therefore obtain a very high loss ($\sim 0.8$), while the last steps contain virtually no noise, and the MSE is very low ($\sim 1e^{-3}$). Therefore, we compute a baseline score to normalize the scale. We subtract from each score the denoising score for the same images using a *random token* which serves as an upper bound for the MSE. Fig. 7(a) presents the results averaged across all concepts, showing that the concept $w^c$ obtains a score worse than both $w^*$ and the optimized token $w^o$, which obtains the best results. These differences are statistically significant, as shown by the error bars marked on every timestep. Evidently, by optimizing a token over a larger domain, we can outperform the original concept token in the denoising task. Importantly, this motivates the different features learned by $w^*$ while also demonstrating the remarkable denoising capability of $w^*$ on *any concept image*, which indicates that $w^*$ indeed captures the concept features.

Fig. 7(b) provides a qualitative comparison between $w^c$ and $w^*$. An input image $I$ generated by *"a photo of a nurse"* is noised and then denoised back from different denoising steps, using the concept token $w^c$ and our pseudo-token $w^*$. As can be seen, there are cases where, given a different random

---

*Our code does not employ a fixed random seed, thus each run implies a different random initialization for the MLP and a different set of random noises and timesteps for the training images.

seed, $w^c$ does not preserve the features in the original image $I$ (*e.g.*, it adds hats, face masks, and black and white effects), while $w^*$ does. Intuitively, this can be attributed to the rich representation learned by $w^*$, which can include both semantic and style features. Both experiments motivate the diversity of the learned decomposition. Since $w^c$ is not necessarily optimal for Eq. 1, $w^*$ learns additional features to improve the denoising quality. Thus, $w^*$ balances two objectives– interpretability and faithfulness to the model's internal representations.

# D  SINGLE-IMAGE DECOMPOSITION

In this section, we provide additional examples of single-image decompositions obtained by our method, as described in Sec. 3 of the main paper. Fig. 8 presents the obtained decompositions for images generated by Stable Diffusion for a given concept. As can be seen, the phenomena demonstrated in the main paper are reproduced in the presented decompositions. For example, *"sweet peppers"* borrow the appearance from the *"pepper"* and the shape from the *"fingers"*, the *"frog"* borrows the overall appearance of a *"toad"* with the color of *"Kermit"*, *etc*. Other concepts are constructed as a composition of related elements, for example, *"happiness"* is constructed as *"children"* who are *"laughing"*, and a *"lamp"* is a *"chandelier"* with a *"bulb"* in it.

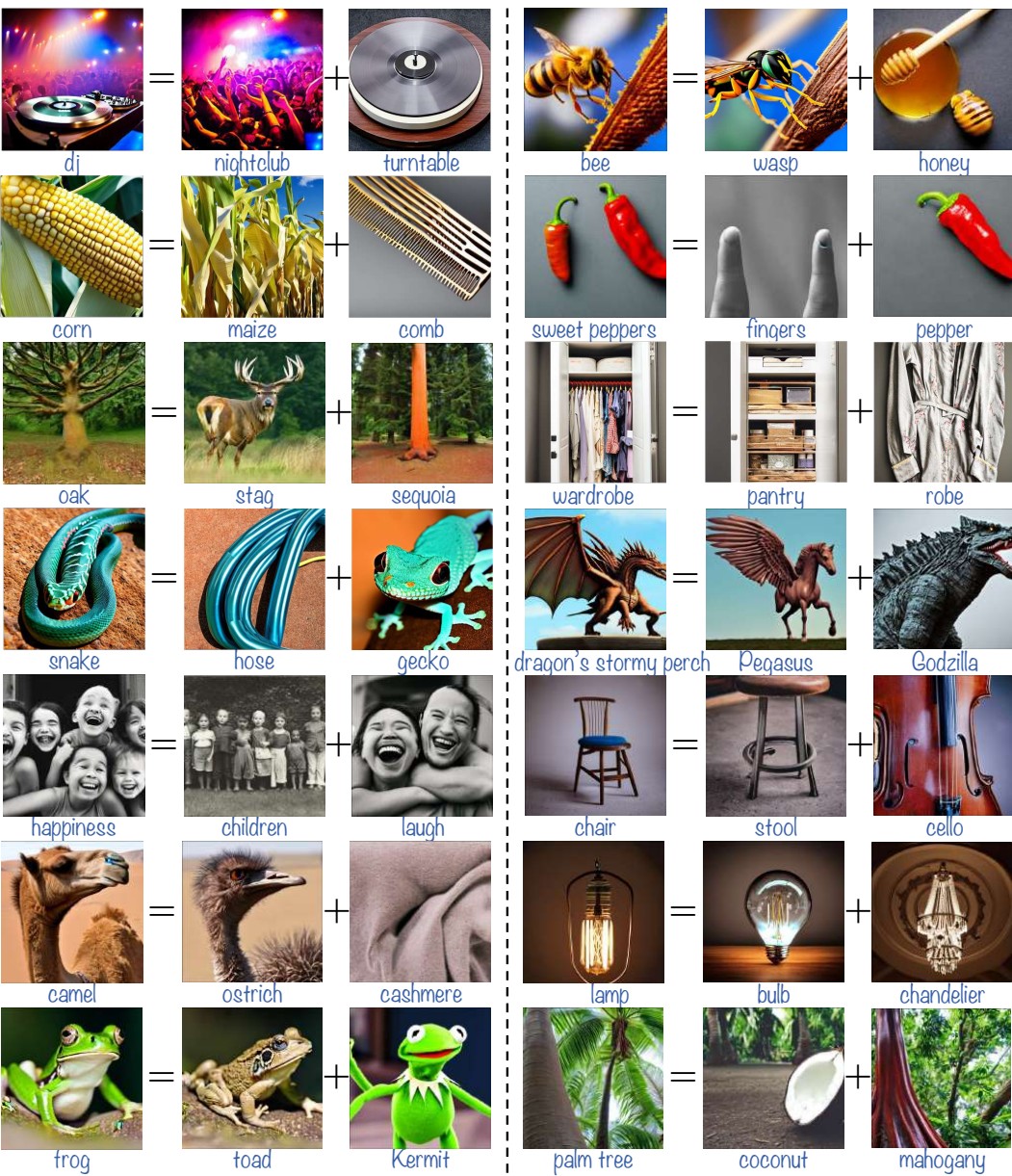

Figure 8: Decompositions of single images by CONCEPTOR. Each of the examples depicts an image generated by Stable Diffusion for the concept, and its corresponding decomposition.

### D.1 RELIANCE ON RENOWNED ARTISTIC STYLES

As demonstrated in Figs. 1 and 2, we find that the representation of the concept *"painter"* relies on the renowned artistic styles of famous artists. In this section, we further verify this observation through the lens of our single-image decomposition scheme. We generate 100 test images for the concept *"painter"* and apply our single-image decomposition scheme to all images. Out of the tested images, we found that 67 images contain at least one name of a famous artist in their decomposition. This result empirically demonstrates the reliance on existing artistic styles, even *when the prompt does not specify the artist's name explicitly*. Fig. 9 demonstrates the impact of the experiment described above on the generated images. We observe that removing the names of renowned artists modifies the painting generated in the image entirely (top row) or, in some cases, removes it from the image altogether (bottom row).

Original Images             w/o Renowned Painters

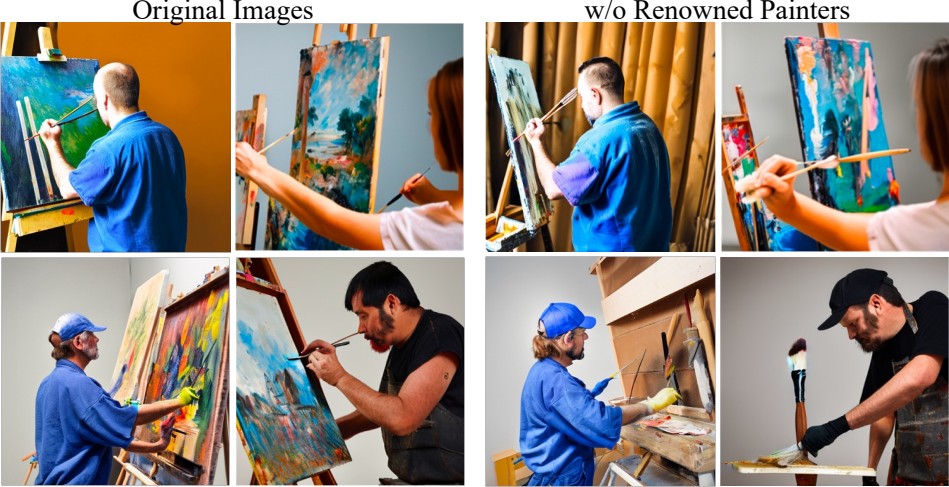

Figure 9: Generated images for the concept *"painter"* before and after removing the names of famous painters from the single-image decomposition. As can be observed, this removal results in a significant modification to the painting in the image.

# E    REPRESENTATION BY EXEMPLARS

Tab. 6 and Fig. 10 present examples of concepts that rely on famous instances for their representations. For example, *"movie star"* is represented by names of famous actors such as Marilyn Monroe or Lucille Ball. Similarly, the concept *"president"* is dominated by American presidents such as Obama and Biden, and the concept *"basketball player"* relies on famous players such as Kobe Bryant and LeBron James. To demonstrate the reliance on the exemplars in the decomposition, Fig. 10 shows the reconstruction by our method with and without the famous instance names from Tab. 6. As can be observed, the reconstruction quality heavily relies on the identities of the instances, and when we remove those instances the reconstruction is harmed significantly.

Table 6: Exemplar-based decomposition elements obtained by CONCEPTOR.

| Concept | CONCEPTOR |
|---|---|
| Composer | Schubert, Beethoven, Chopin, Mozart, Brahms, Wagner |
| Movie Star | Aubrey, Bourne, Lucille, Gloria, Marilyn, Monroe, Oswald |
| President | Obama, Trump, Biden, Nixon, Lincoln, Clinton, Washington |
| Rapper | Tupac, Drake, Khalifa, Weekend, Khaled, Eminem, Wayne |
| Basketball player | Kobe, LeBron, Shaq, Bryant, Jordan, Donovan, Kyrie |

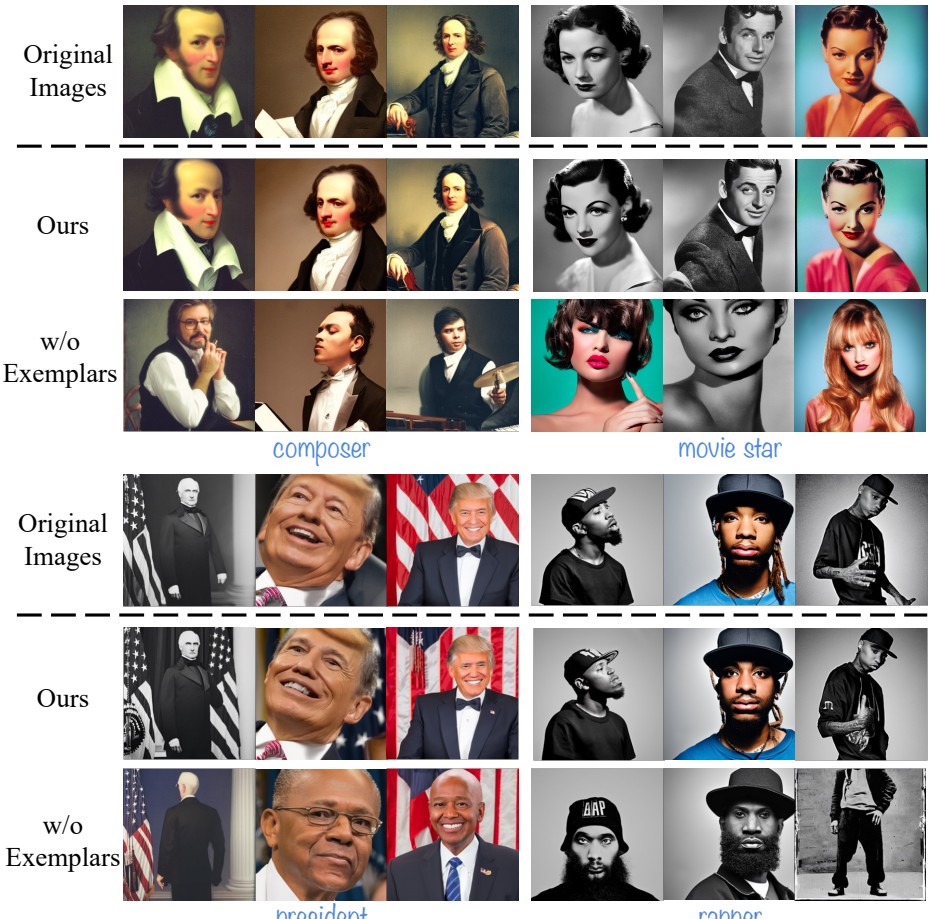

Figure 10: Examples of concepts that rely on famous instances. When removing the exemplars, the reconstruction quality is significantly harmed.

## F DECOMPOSITION OF ABSTRACT AND COMPLEX CONCEPTS

In Fig. 11 we present decomposition results for abstract and complex prompts from our dataset (see list in Appendix B) in the form of word clouds. As can be observed, our method successfully produces meaningful decompositions for various abstract concepts including emotions (*e.g.*, *"happiness"*, *"sadness"*) as well as complex abstract concepts (*e.g.* *"elegance on a plate"*). The decompositions demonstrate intriguing connections between abstract elements (e.g., *"happiness"* is linked to *"childhood"*, *"laughter"*, *"dream"*).

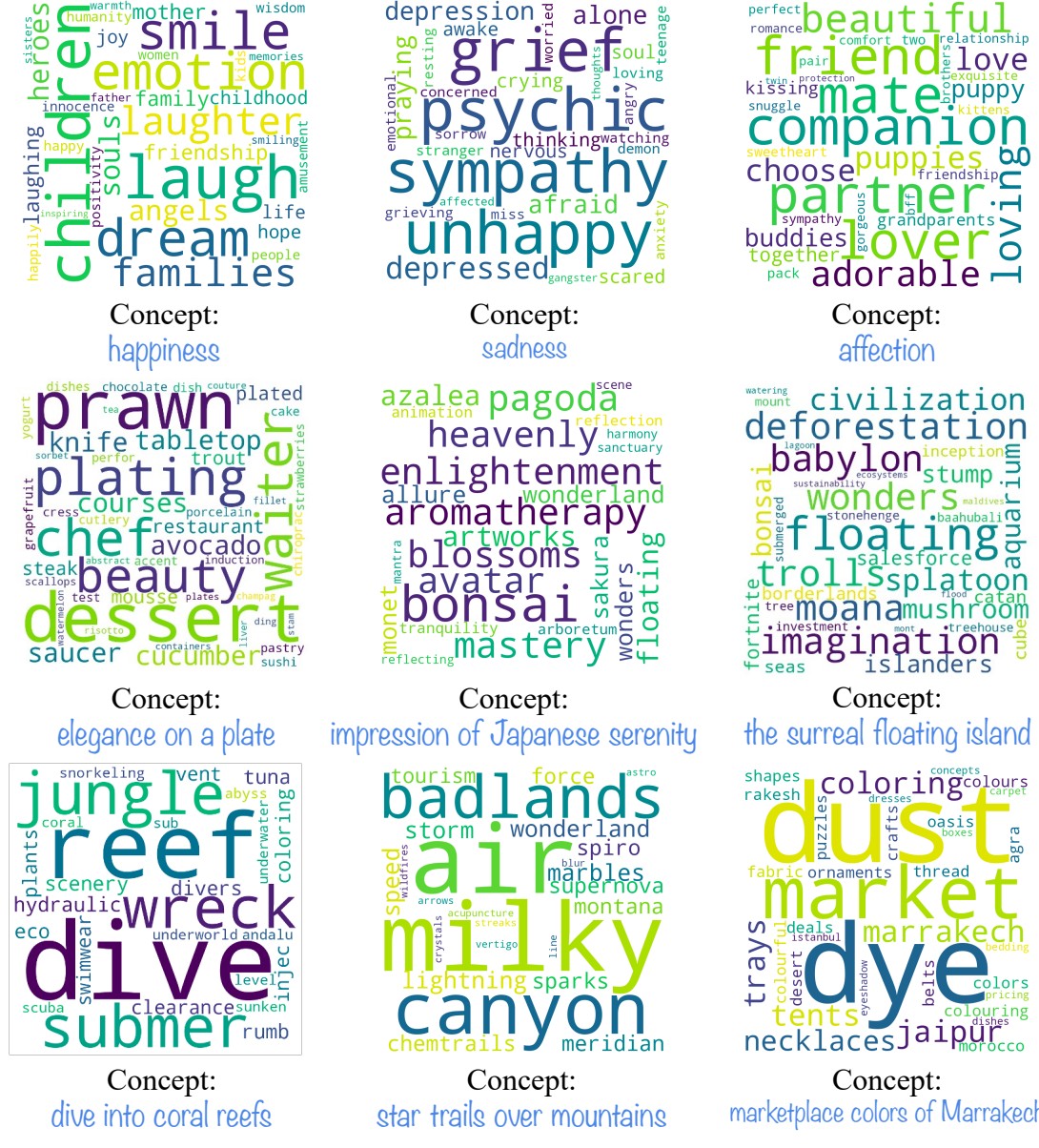

Figure 11: Decompositions obtained by CONCEPTOR for abstract and complex concepts from our dataset.

# G  OBJECT-CENTRIC REPRESENTATION METHODS

The closest relation we observed between object-centric representation methods and CONCEPTOR is through the task of unsupervised object/ concept discovery. Importantly, the task of interpreting diffusion models largely differs from that of object discovery, since an interpretation should often contain elements *beyond* objects that are physically present in the image. For example, Fig. 2 shows connections such as *"snake"* to *"gecko" + "twisted"*, a *"gecko"* is not present in the image, and *"twisted"* is not an object but an adjective. Similarly, *"cashmere"* is not present in an image of a *"camel"*, *etc*. In contrast to these semantic, profound connections, interpretation using object discovery only provides representations that are based on concrete objects and parts that are visible in the images and falls short of discovering the additional connections required to interpret the model (*e.g.*, reliance on exemplars, non-trivial biases, *etc*.).

To empirically demonstrate this point, Fig. 12 presents exemplary results using (Liu et al., 2023), which is the state-of-the-art concept discovery method based on SD. Fig. 12 performs concept discovery of three exemplary concepts from our dataset, *"snail"* and *"snake"*, and a complex concept *"impression of Japanese serenity"*. As can be seen (first column for *"snail"* and *"snake"*, fourth for *"impression of Japanese serenity"*), the method learns to embed the concept in a single token (as expected from its task definition), *even if the concept is complex*, and does not employ any other learned tokens, therefore they are semantically meaningless for interpretability purposes.

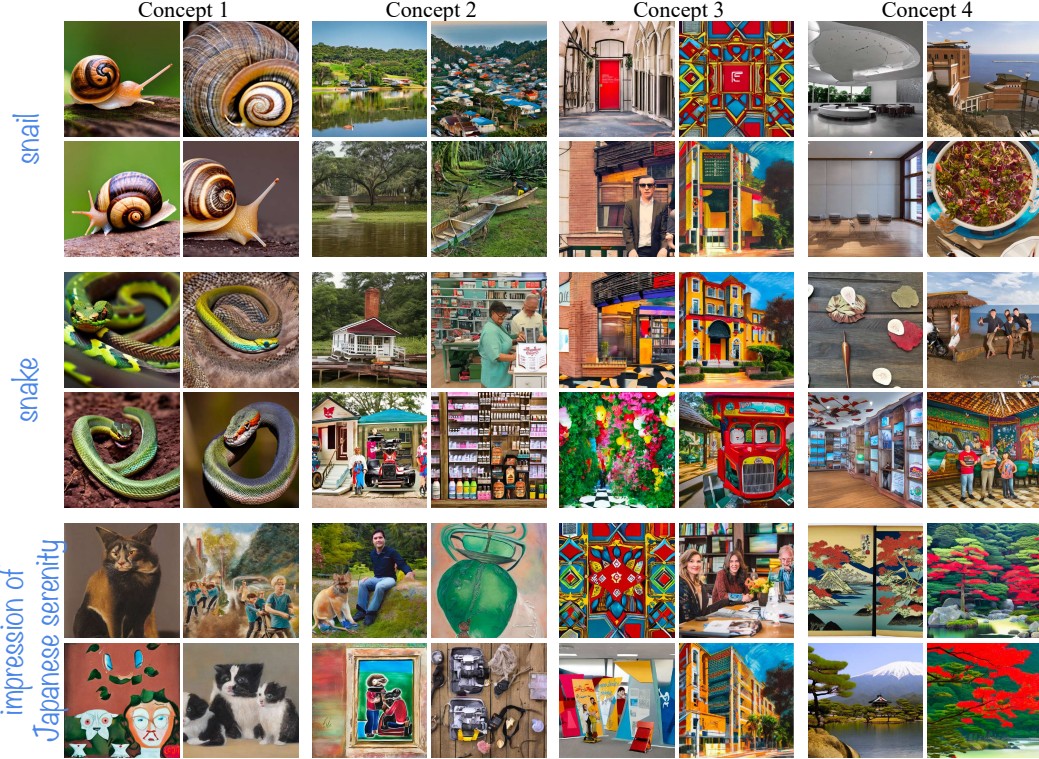

Figure 12:  Examples of concept discovery with (Liu et al., 2023) for *"snail"*, *"snake"*, and *"impression of Japanese serenity"*. As can be observed, the entire prompt is encapsulated in a single concept, while the other discovered concepts are random. This aligns with the objective of the method, to divide all concepts in the image between the different tokens.

## H CONCEPT DEBIASING

As mentioned in the main paper, CONCEPTOR is capable of detecting biases that are otherwise difficult to capture by simply observing the generated images. After detecting a bias, one could employ coefficient manipulation with our method to mitigate the bias. Fig. 13 presents examples of such concept debiasing for the concepts in Tab. 7. For each concept (row), we decrease the coefficient of the biased tokens in the decomposition until an unbiased representation is achieved. Fig. 13 demonstrates a comparison of the generated images by SD without intervention (Original Images), and the images reconstructed by our method after reducing the bias coefficients, on the same set of 8 random seeds. As can be observed, our method is able to mitigate the different biases while maintaining the other features represented by the concept.

Table 7: Biased terms detected and removed by CONCEPTOR using coefficient manipulation.

| Concept | CONCEPTOR |
|---|---|
| Professor | `men` |
| Nurse | `woman, women, wife, mother` |
| Secretary | `wife, hostess, womens, girl, ladies` |
| Opera singer | `obese, overweight, fat` |

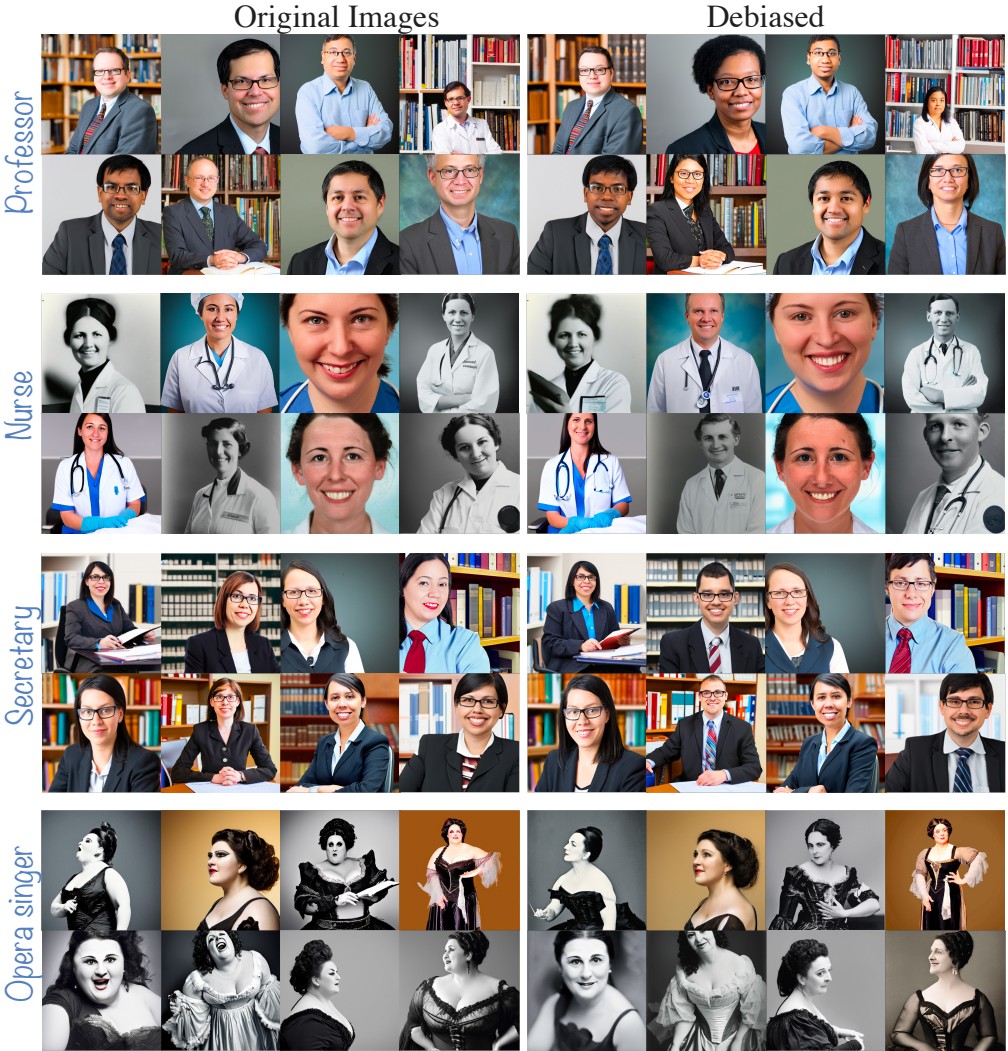

Figure 13: Examples of concept debiasing using CONCEPTOR with the same set of 8 random seeds. The first three examples demonstrate gender debiasing, while the last one demonstrates a decoupling between *"opera singer"* and `obesity`.

# I    Principal Component Visualization for Baselines

In this section, we enclose the concept activation vectors (CAVs) learned by the leading concept-based explainability method adapted from CNNs (PCA). Fig. 14 present the reconstruction and components learned by PCA. To visualize a component, we remove it and generate the corresponding images without it to examine its impact on the generations. As can be seen, most of the learned components are not easily interpretable and do not demonstrate a coherent semantic change across the different test images. Thus, these methods violate the *meaningfulness* criterion. We note that meaningfulness is a critical criterion for an interpretation, as without it, humans cannot understand the decomposition, and it does not serve its basic goal of giving insights into the model.

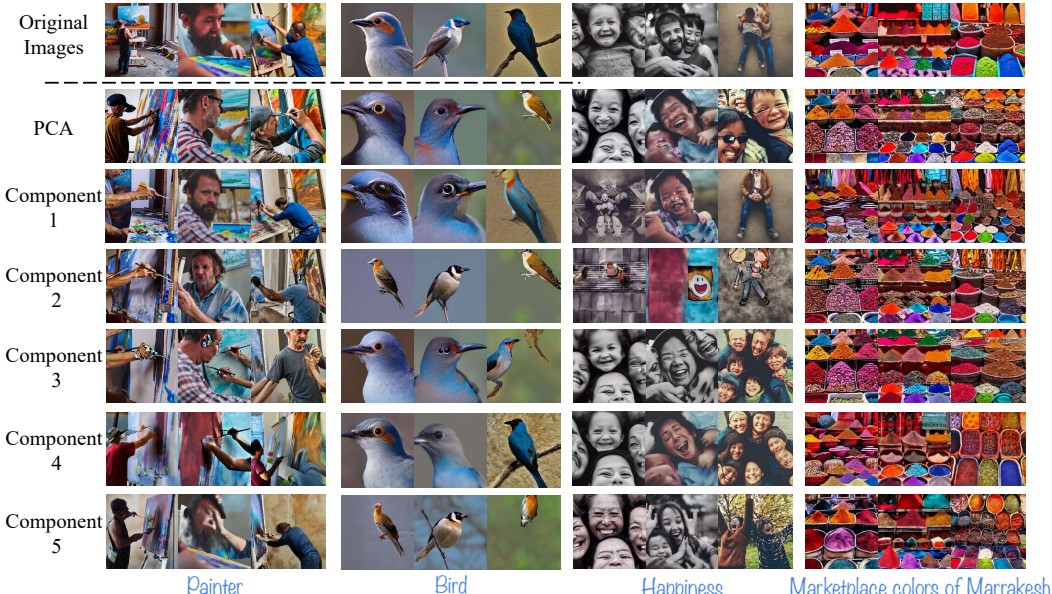

Figure 14: PCA top 5 extracted principal components for 4 concepts. The first row depicts SD's original images, the second row shows the reconstruction by PCA, and the last 5 rows demonstrate the impact of removing each of the top 5 principal components learned by PCA.

## J    OUT OF DOMAIN (OOD) EXPERIMENTS

The main objective of our method is *the interpretation of the inner representations of concepts by the model*. With that, in this section, we explore CONCEPTOR's ability to provide insight into how the model represents out-of-domain concepts.

We present examples of three out-of-domain concepts extracted from the DreamBooth dataset (Ruiz et al., 2022). These examples represent increasing distance from the model's distribution. The first concept represents an example of a concept that is close to the model's distribution, *i.e.*, a specific dog. The second one deviates further from the distribution and portrays a sloth plush. Finally, we present an example of a concept that is completely out-of-domain, a monster doll with a unique appearance that the model is not familiar with.

We use CONCEPTOR to interpret the representation of each of these concepts and apply a baseline where we mimic CONCEPTOR's optimization without any restrictions on $w^*$ (similar to (Gal et al., 2022)). The results for all three concepts are presented in Figs. 15 to 17, respectively.

As can be expected, the reconstruction quality of both the baseline and CONCEPTOR degrades as we move further away from the model's distribution. However, interestingly, even in considerable OOD cases (*i.e.*, the sloth plush, the monster toy) the decompositions by CONCEPTOR remain meaningful, as is reflected by the decomposition elements presented in the bottom two rows of each figure. For example, the sloth plush in Fig. 16 is linked to the concepts *"plush"*, *"fluffy"*, and *"knitted"*, and the monster toy in Fig. 17 is linked to *"woolly"* due to its wool-like fur, to the character *"Domo"* due to the shape of its body, and to a *"crab"* and a *"shrimp"* due to its unique limbs. These results further substantiate our method's powerful ability to learn meaningful decompositions, even in the face of OOD samples.

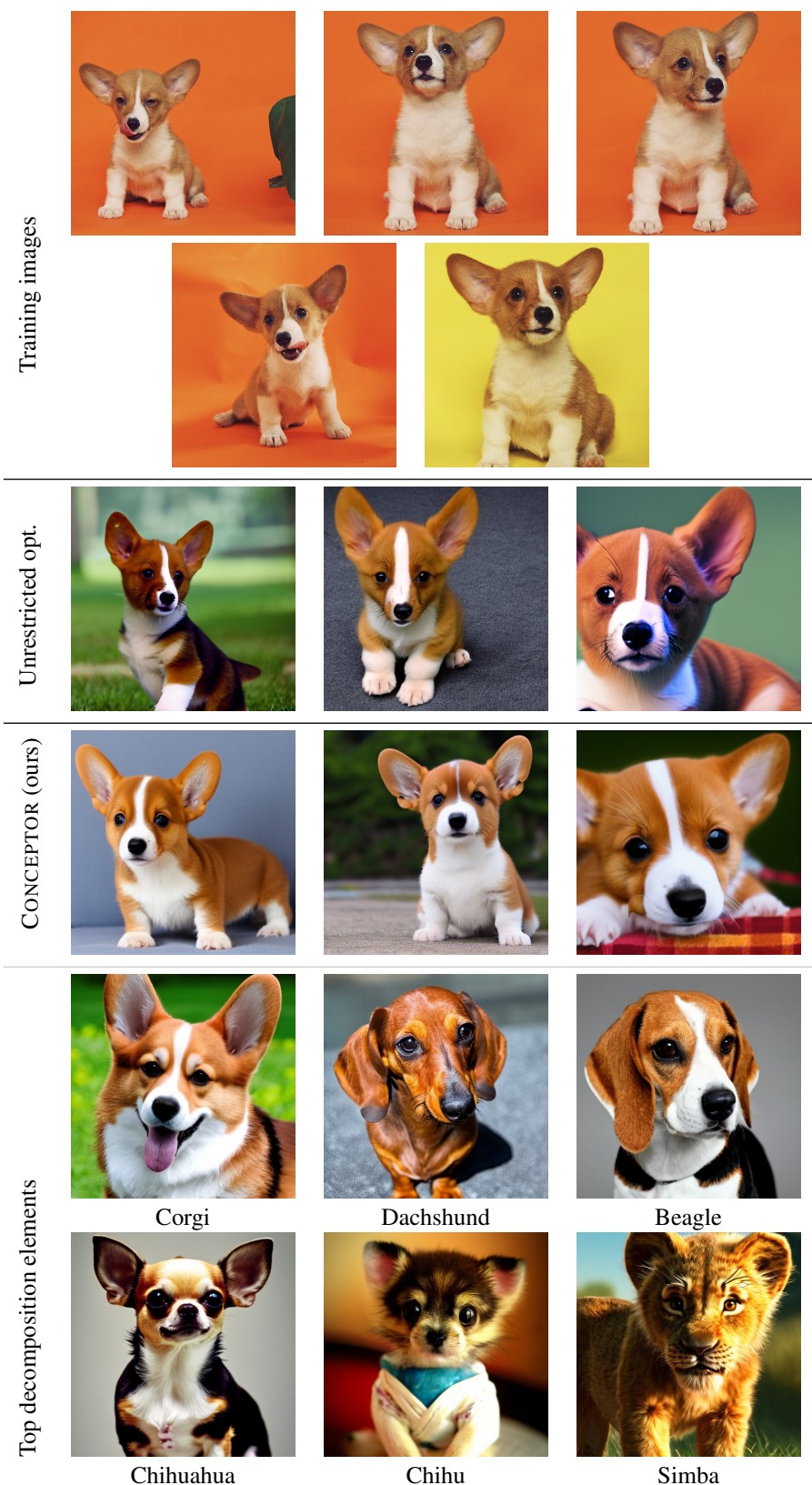

Figure 15: Out of domain samples. The first two rows depict the training images from Ruiz et al. (2022). The next row depicts samples drawn from the concept learned by an unrestricted optimization. The third row depicts samples created from the concept we learned, $w^*$. The last two rows present the primary decomposition elements learned by our method for the concept.

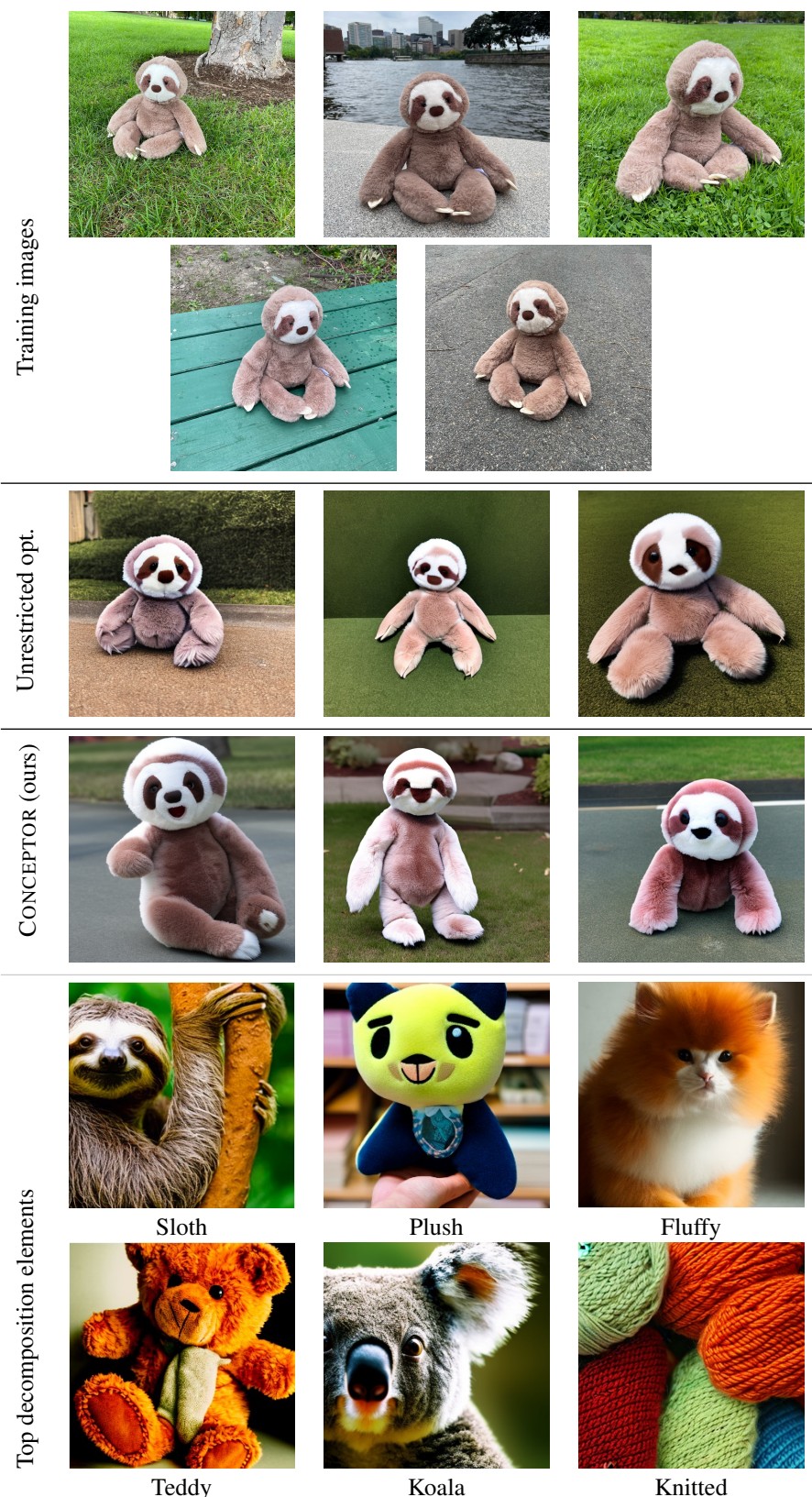

Figure 16: Out of domain samples. The first two rows depict the training images from Ruiz et al. (2022). The next row depicts samples drawn from the concept learned by an unrestricted optimization. The third row depicts samples created from the concept we learned, $w^*$. The last two rows present the primary decomposition elements learned by our method for the concept.

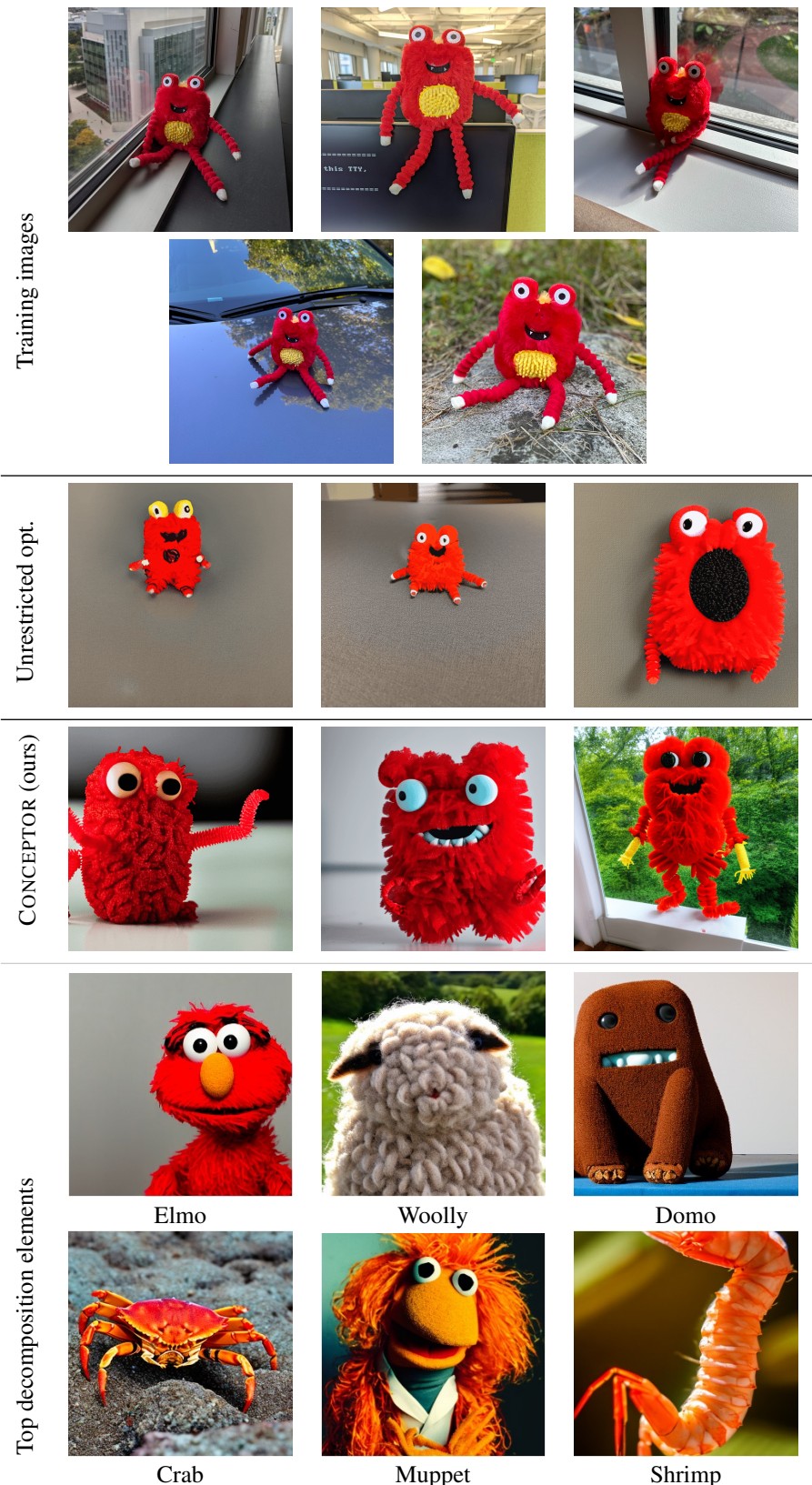

Figure 17: Out of domain samples. The first two rows depict the training images from Ruiz et al. (2022). The next row depicts samples drawn from the concept learned by an unrestricted optimization. The third row depicts samples created from the concept we learned, $w^*$. The last two rows present the primary decomposition elements learned by our method for the concept.

## K   ROBUSTNESS TO ALTERNATIVE LOSS FUNCTIONS

In our experiment, we chose to employ the model's reconstruction loss (Eq. 1) to encourage our decomposition to encapsulate the features learned by the model for the concept. In this section, we explore the possibility of replacing the MSE reconstruction with a semantic similarity engine, *i.e.*, CLIP (Radford et al., 2021). Since the UNet, $\varepsilon_\theta$, predicts the added noise $\epsilon$ (for which CLIP is not directly applicable), we apply our loss over the prediction of the clean image, $\hat{x}_0$, *i.e.*:

$$\mathcal{L}_{CLIP} = 1 - cosine\left(CLIP(x_0), CLIP(\hat{x}_0)\right), \tag{7}$$

where $x_0$ is the clean training image, and $\hat{x}_0$ is the predicted image by the diffusion model, *i.e.*:

$$\hat{x}_0 = \frac{x_t - \sqrt{1 - \bar{\alpha}_t}\hat{\epsilon}}{\bar{\alpha}_t}, \tag{8}$$

where $x_t$ is the noised version of the input image $x_0$ according to Eq. 2, and $\hat{\epsilon}$ is the noise predicted by the UNet. In other words, we calculate the predicted clean image based on the noise prediction and apply the CLIP loss against the original image. This process will result in a decomposition that gives us the elements best correlated with the features of the concept *according to CLIP*. We note that since CLIP is a semantic engine, some degradation in the reconstruction performance is to be expected. In Tab. 8 and Fig. 18 we present a qualitative comparison between the decomposition and reconstruction obtained by our method with the CLIP loss, compared to the original MSE loss used. We employ the same concepts and seeds from Fig. 4 of the main paper. As can be observed, while the reconstruction quality indeed decreases, the same phenomena observed with the MSE loss are reproduced with the CLIP loss. For example, the concept *"painter"* still heavily relies on *the same* famous artists discovered by our full method (*e.g.*, *"Monet"*, *"Picasso"*). These results further substantiate the conclusions presented in the main paper, since it is shown that the method is insensitive to the choice of the reconstruction loss.

Table 8: A comparison between the decompositions obtained with CONCEPTOR with MSE loss vs. CLIP image loss.

| Concept | CONCEPTOR w/ CLIP Loss | CONCEPTOR w/ MSE Loss |
|---|---|---|
| Painter | Monet, painter, Picasso, watercolor, Impressionist, photographing, painting... | Monet, Picasso, paint, Impressionism, artist, studio, brushes, sketching... |
| Bird | kingfisher, bunting, bluebird, starling, magpie, finch, parrot, jays... | teal, beak, brown, stripped, dove, quail, sparrow, hummingbird, finch... |
| Happiness | smile, warmth, happiness, children, delight, emotion, cheerful, love, friendship... | children, smile, laughter, dream, families, emotion, smile, mother, friendship... |
| Marketplace colors of Marrakech | flavor, Bangkok, Mumbai, souvenir, fragrances, flavors, tourist, bazaar, spices... | Marrakech, tents, coloring, market, fabric, desert, oasis, spices, dust, ornaments... |

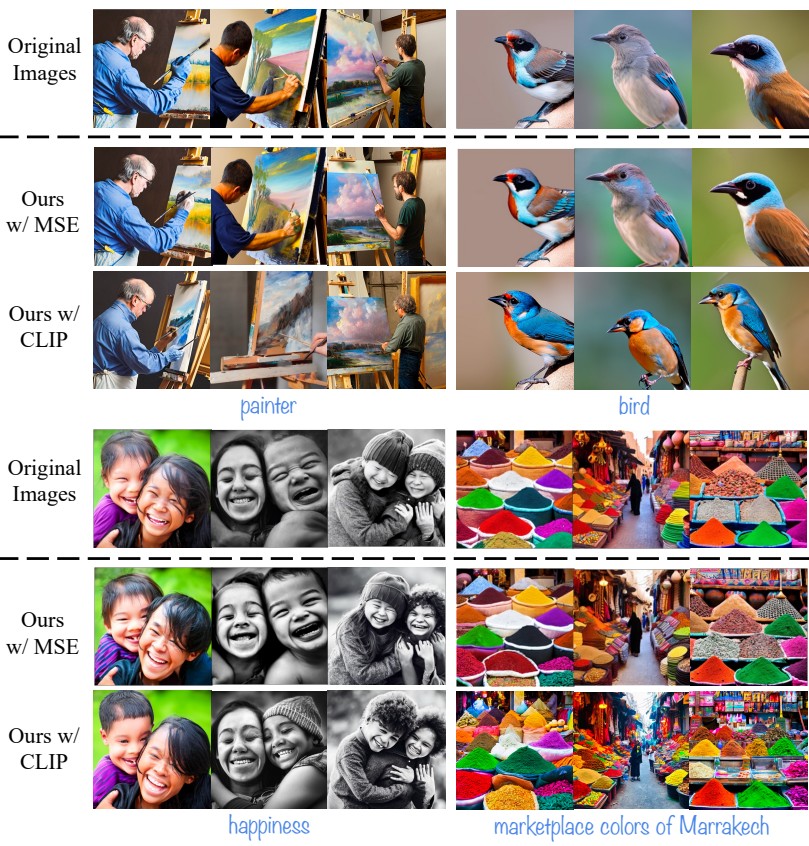

Figure 18: Feature reconstruction comparison between our method with an MSE loss and with CLIP loss. The original images appear on the first row.

## L ROBUSTNESS TO THE MODEL SELECTION

In this section, we conduct an experiment to demonstrate that CONCEPTOR is robust to the choice of the text encoder and the model. We apply our method over SD 1.4, which employs a different text encoder than that of SD 2.1. Importantly, SD 1.4 uses CLIP ViT-L/14 which was trained on proprietary non-public data, while SD 2.1 uses OpenCLIP-ViT/H, which is an open-sourced version of CLIP, trained on public data. These two text encoders are entirely different not only in training data but also in the size of their word embedding vectors (768 for CLIP ViT-L/14 vs. 1024 for OpenCLIP-ViT/H). We compute the decompositions by CONCEPTOR for the same concepts as those in Fig. 4, in the same setting described in the paper. The qualitative results, both the textual decompositions and the feature reconstruction results, are presented in Tab. 9 and Fig. 19, respectively. As can be seen, the decompositions remain human-understandable and faithful to the model, as expected. Additionally, to establish that reliance on exemplars exists across versions of SD (independently of the training data and text encoder), we enclose in Tab. 9 the decomposition for the concept *"president"*, showing that it is still dominated by former American presidents.

Table 9: Textual decomposition elements obtained by CONCEPTOR for SD 4.1.

| Concept | CONCEPTOR |
|---|---|
| Painter | painting, watercolor, sketch, studio, pastels, artist, canvas, ladders, oilpainting, brush... |
| Bird | avian, birdie, jay, phoebe, pilgrim, blackbird, hummingbird, whistler, peregrine, cardinals... |
| Happiness | friendships, childrens, rejoice, love, photography, unforgettable, everyone, dreamers, smiles... |
| Marketplace colors of Marrakech | marketplace, marrakech, paprika, rugs, terrace, dyes, vendors, barrels, bowls, colorful... |
| President | potus, barackobama, obama, nixon, washington, realdonaldtrump, jefferson, washington, roosevelt... |

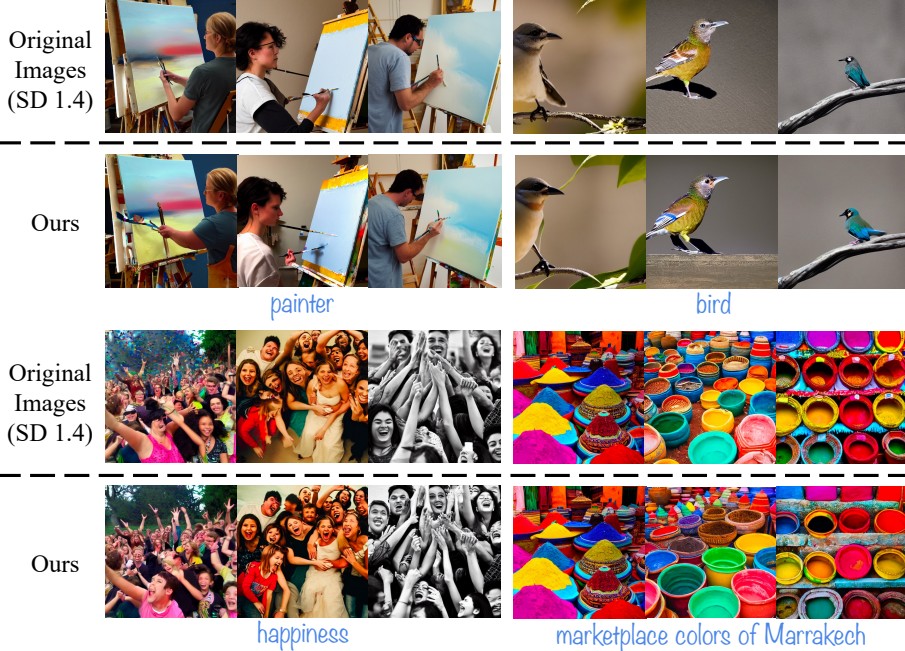

Figure 19: Feature reconstruction results of our method on SD 1.4.

