# OpenReview forum: "The Hidden Language of Diffusion Models"
_ICLR.cc/2024/Conference — ICLR 2024 poster_

### Official Review · Reviewer_vrPv · 2023-10-29

**Soundness:** 3 good
**Presentation:** 4 excellent
**Contribution:** 2 fair
**Rating:** 6
**Confidence:** 5

**Summary:**

The paper proposes to use a weighted linear combination of existing word embeddings to represent an image (i.e., weights are optimized through two MLP layers) such that it enables image decomposition based on a set of human understandable tokens using pretrained text-to-image diffusion model such as Stable diffusion models.

**Strengths:**

- the motivation of the paper is clear and well conveyed.
- the experiments are well-conducted and extensive.
- The approach enables human interpretable decomposition using a set of learnable weights and associated tokens, which has inherited the same outcome from the classic word2vec arithmetic paper [1].

[1] Mikolov et al., Efficient Estimation of Word Representations in Vector Space. ICLR 2013

**Weaknesses:**

- lack of related previous work on image decomposition, which can be seen as a way to interpret models, such as FineGAN (Sing et al, 2019), GIRAFFE (Niemeyer etal, 2021), SlotAttention (Locatello etal, 2020), DTI Sprites (Monnier etal, 2021), GENESIS-V2 (Engelcke etal, 2021) and follow-up works. There also exists earlier/concurrent work that conducts image decomposition using text-diffusion models, so its worth discussing pros and cons but may not need comparison if works are concurrent.
- most of concepts shown in the paper are mainly objects, thus its ability to learn abstract concepts is not clear. For example, how does it perform on abstract concepts such as object relationships. Though, I tend to think that the model is rather limited in understanding complex concepts other than objects.
- The method essentially utilizes arithmetic with word embeddings which have been widely used in the past, so it doesn't seem to be novel enough. Applying this method to a text-to-image diffusion model doesn't show novelty from my perspective.

**Questions:**

- I'd love to see if u can optimize on electrician image and then try removing the brush concept, does it become a painter?
- How long does this optimization process take for each image?

---

> ### Author Response · Authors · 2023-11-12
> **Thank you for the review! (response part 1/3)**
>
> We thank the reviewer for the comprehensive feedback and for the interesting points for discussion. We are encouraged that the reviewer found our work to be well-motivated and recognized the extensiveness and effectiveness of our experiments.
> In the following, we provide a response to each of the weaknesses pointed out in the review. Additionally, the modifications in the revision are marked in red, for your convenience.
>
> __Re. Related works on image decomposition:__
> We thank the reviewer for bringing this to our attention. We have revised the paper to include the proposed works in the related works section.
>
> To reiterate the difference between the objective of these methods and our method, we enclose a short comparison to the methods indicated above:
>
> As a general note, all methods described below only offer decomposition for a single image and, therefore, fall short of providing concept-level explanations, which is the main objective of concept-based explainability.
>
> **FineGAN**: disentangles background, foreground, shape, and appearance. This method is not applicable to interpret SD since (1) it’s GAN-based and therefore limited to the training domain, (2) the disentangled information provides little to no interpretable information that is not observable just by looking at the images.
>
> **GIRAFFE**: similar to FineGAN, GIRAFFE disentangles objects, appearance, shape, and background information, with the objective of obtaining controllable generation. In addition to the points mentioned above, this method incorporates 3D scene representations, which are not applicable in our case.
>
>  **Slot Attention**: learns object-centric representations for complex scenes, and employs them for object discovery and set prediction. The task of object discovery has some similarities to ours but is also very different, see below. Additionally, the datasets used for training (CLEVR6, Multi-dSprites, Tetrominoes) differ significantly from the distribution of images generated by SD.
>
> **DTI Sprites**: similar to Slot Attention, DTI Sprites decomposes a scene into objects including their shape, size, and color (appearance), which allows improved controllable generation.
>
> Inspired by the review, we conducted an additional experiment to demonstrate the difference between our objective and existing methods for image decomposition. The closest relation we observed between the proposed methods and ours is through the task of unsupervised object discovery (including shapes, colors, sizes, etc.). Importantly, our task largely differs from that of object discovery, since an interpretation should often contain elements *beyond* objects that are physically present in the image. For example, Fig. 2 shows connections such as “snake” to “gecko” + “twisted”, a “gecko” is not present in the image, and “twisted” is *not an object but an adjective*. Similarly, “cashmere” is not present in an image of a “camel”, and “winding” is not present in images of “snails”. In contrast to these semantic, profound connections, interpretation using object discovery only provides representations that are based on concrete objects and parts that are visible in the image. Therefore, these methods fall short of discovering deeper connections (e.g., reliance on exemplars, non-trivial biases, etc.).
>
> To empirically demonstrate this point, we added exemplary results using [1], which is the state-of-the-art object discovery method based on SD, and allows to discover concepts from sets of images, as well as single images. The results of this exploration of two exemplary basic concepts, “snail”, “snake” and an additional complex concept “impression of Japanese serenity”, are presented in Appendix G of the revision. This experiment directly demonstrates our intuition above. As can be seen, the method learns to embed the concept in a single token (as expected from its task definition for images of one class) and does not employ any other learned tokens.
>
> [1] “Unsupervised Compositional Concepts Discovery with Text-to-Image Generative Models”, Nan Liu, Yilun Du, Shuang Li, Joshua B. Tenenbaum, Antonio Torralba, ICCV’23.

---

> ### Author Response · Authors · 2023-11-12
> **Thank you for the review! (response part 2/3)**
>
> __Re. Forming abstract connections beyond objects:__
> Kindly note that our examined concepts include an extensive set of both very complex and abstract concepts (e.g., “elegance on a plate”, “happiness”, “affection”, “impression of Japanese serenity”, etc.). A full list of these concepts can be found in Appendix B (see Emotions, Actions, Complex concepts). Following the review, we have added word cloud visualizations of the learned decompositions for abstract and complex concepts (see Appendix F). The provided examples show that our method can meaningfully decompose abstract concepts and detect relations to other abstract concepts (e.g., linking “happiness” to “dream”, “emotion”, “soul”, “laughter”, “childhood”; or linking “sadness” to “grief”, “depression”, “worried”, “alone” etc.). These results indicate that Conceptor is capable of revealing connections between abstract concepts.  Examples of single-image decompositions for such abstract concepts can be found in Fig. 2 (“fear”, “impression of Japanese serenity”) and in Fig. 8 (“happiness”).
>
> __Re. The difference with word arithmetics:__
> We thank the reviewer for raising this interesting point for discussion.
> We wish to emphasize that our method does not perform word embedding decomposition, and in fact, significantly differs from it. In the following, we list the fundamental differences between the two:
>
> 1. **Our learned pseudo-token $w^{\ast}$ does not decompose the text embedding of the concept $w^c$, i.e.: $w^{\ast}= \sum_i w_i \alpha_i \neq w^c$:**
>
>       Kindly note that our method does not decompose the word embedding of the concept token(s) $w^c$. Instead, we propose to decompose the internal representation of the concept by the model. This is done by aggregating the set of features used by the model in the image generation process (see Sec. 3). The text embeddings serve merely as an intermediate language linking us to the uninterpretable inner representations of SD. To empirically prove this point, we calculated the cosine similarity between our learned pseudo-token $w^*$ and the concept token $w^c$ averaged across our entire dataset, and obtained a low similarity score of *0.61* (for reference, “cat” and “car” get a higher similarity score of 0.66). This result, combined with our Token Diversity metric (Tab. 1), our single image decomposition results (Fig. 2) and our CLIP top words ablation (Tab. 3) demonstrate that our method is able to learn connections based on visual semantics, that transcend word arithmetics, as mentioned in the paper (see Sec. 1, 6).
>
> 1. **Our single-image decomposition results do not suggest word embedding arithmetics:**
>
>      Note that the annotations in our single-image decomposition figures (Figs. 2,8) do not indicate word embedding arithmetics. For example: $word\textunderscore embed(painter) \neq word\textunderscore embed(electrician) + word\textunderscore embed(brush) + word\textunderscore embed(Monet)$ (the cosine similarity between “painter” and “electrician + brush + Monet” is merely 0.496). This indicates that the decomposition is not based on textual semantics but on an inner-representation decomposition.
>
>       Additionally, note that a diffusion model generates *diverse* images for the same concept given different random seeds, and not all learned features are manifested in all of the concept images (e.g., the images of the nurses in Fig. 12 can be in black and white or in color, with scrubs or a white suit, with and without a stethoscope, etc). Our single-image decomposition scheme extracts the specific set of elements from the decomposition learned by Conceptor that are used to generate a single specific image. This is evident when different features appear for single-image decompositions of different images of the same concept, e.g., the “camel” and “snake” examples in Fig. 2 vs. Fig. 8. Thus, the arithmetic annotations in Figs. 2, 8 indicate the addition of the features of the decomposition elements to the output image.
>
>     This ties directly to the question by the reviewer:
>      > “I'd love to see if u can optimize on electrician image and then try removing the brush concept, does it become a painter?”.
>
>      The answer is that we are not doing word arithmetics and this manipulation is not possible, since applying Conceptor to the concept “electrician” results in a decomposition that does not include the element “brush”. This is logical: SD does not employ the feature “brush” in the generation of images of electricians, unlike the role “brush” plays for the “painter” concept.

---

> ### Author Response · Authors · 2023-11-12
> **Thank you for the review! (response part 3/3)**
>
> __Re: Novelty:__ As mentioned and demonstrated above, our novel decomposition formulation has not been attempted before and differs significantly from other image decomposition objectives such as object/concept discovery. As we established, Conceptor also fundamentally differs from word embedding arithmetics.
>
> Our experiments demonstrate that Conceptor has significant value in exposing mixed visual/textual components in a way that, as far as we can ascertain, has never been shown before. For example, Conceptor exposes reliance on exemplars and mimicking of artistic styles and reveals concepts that are incorporated purely for visual reasons such as shape or texture, non-trivial biases, and the way textual ambiguities are handled.
>
> From a technical point of view, we believe that there is a considerable novelty in our decomposition strategy, e.g., in the combination of Eqs. 3 and 4, in which *the coefficients $\alpha$ are modeled as a function of the word embedding $w$* (see Sec. 3). These unique choices are extensively ablated in Sec. 4.2.1, where we empirically demonstrate that they are all crucial to maintaining the faithfulness and meaningfulness of our method. Our single-image decomposition scheme introduces additional innovation by enabling interpretation for a specific generation, in addition to the general concept interpretation provided by our method.
>
> __Re. Optimization length per image:__ Please note that Conceptor *does not operate on a single image but on an entire concept* (see Sec. 3). The optimization process for the entire concept takes around 6.5 minutes on a single A100 GPU with 40GB (given more memory, one could significantly increase the batch size and speed up the process).
>
> After this optimization, manipulation of the obtained coefficients for a given image is dominated by the time it takes to generate the image (around 4 seconds, again depending on the hardware used).
>
> We are happy to address any other questions.

---

> > ### Comment · Reviewer_vrPv · 2023-11-19
> >
> > 1. Thank you for adding image decomposition to the related work and the additional baseline to showcase your method.
> > 2. The main reason why I mentioned "word arithmetics" is because: since you select a set of $n$ tokens to optimize, such $n$ tokens will be in a vector subspace $\leq N^m$ where $m <= n$. As a result, no matter how you optimize it, it will be always in that subspace. Suppose they lie in the subspace of $N^m$  and the linearly independent vectors that make up that subspace are $c_1, \dots, c_n$, then your final $w^{*}$ will always be some weighted linear combination of $c_1, \dots, c_n$. In that case, your method is essentially word arithmetics but with learnable parameters from my interpretation, though $c_1, \dots, c_n$ may not human interpretable concepts. After parameters are learned, it seems to be exactly word arithmetics in that sense in test time where you use "+/-" to add or remove concepts. Please correct me if I am wrong or misunderstood it.
> > 3. Thank you for addressing other questions and concerns. Can't think of any questions right now, but will come back for questions if I do have any additional ones.

---

> ### Author Response · Authors · 2023-11-19
> **Authors' response**
>
> Thank you for your response to our rebuttal! We appreciate the continued discussion.
>
> In addressing the second point, the literature on word arithmetics shares with our method the use of a vocabulary in which each word is embedded as a vector in some vector space. However, the two lines of research diverge at this point, and it is crucial for us to emphasize the novelty of our method:
>
>
> 1. Unlike the goal of word embedding arithmetics, we do not measure distances in the vector space. Specifically, our decomposition of $w^c$ is not intended to find a weighted sum of dictionary elements equal to it. Instead, we measure the reconstruction loss of generated images.
>
> 2. Our method does not optimize the set of coefficients directly. Instead, it operates by learning a mapping function between each word embedding and a corresponding coefficient. In other words, our method can be viewed as a *nonlinear mapping* of the form: $w^* = \sum_i f(w_i) w_i$, where $f$ is a nonlinear function. Again, this is different from word arithmetics, which considers linear operations in the vector space.
>
>     Our extensive ablations (Sec. 4.2.1, Tab. 3 “without MLP”) show that this novel form of learning the decomposition is critical to the success of the method, as without it (i.e., learning separate coefficients) the decompositions are not faithful. This ablation shows that the task of assigning the appropriate coefficients to word embeddings to reconstruct the learned representation is not trivial.
>
> 3. As a result of 1+2, the tokens obtain their meaning from the context of the decomposition and not from their single-word meaning (see also Fig. 5). For example, as a standalone token, “winding” generates an image of winding roads and spirals, however in the context of the decomposition for “snail” its meaning is transformed to the shape of the snail’s shell. “Civilization” paired with “sphere” turns a simple sphere into planet Earth, etc.
>
>      To further substantiate this point empirically, we enclose the results of simple word arithmetics (a+b) for examples of concepts from Fig 2. The connection “reflections of earth = sphere + civilization” is only made possible by the learned coefficients. When considering only the word embeddings, [these are the resulting images](https://i.imgur.com/6ai71Ww.jpg) (which do not resemble planet Earth whatsoever). Similarly, “snake = twisted + gecko” would yield [these images without the appropriate coefficients](https://i.imgur.com/1UYOham.jpg), for the combination “snail = winding + ladybug” [these are the resulting images](https://i.imgur.com/EB4S5ya.jpg), etc.
>
> 4. Finally, we would like to point out that there are many other lines of research, such as dictionary learning, that learn a combination over a set of dictionary vectors. Therefore, we do not believe that the fact that we are optimizing a combination of tokens is enough to classify our work as a form of word embedding arithmetics. As we showed above, our method significantly differs from word embedding arithmetics in its goal, implementation, and objective function.
>
>
>
>
> Overall, we believe that our obtained results are non-trivial and profound, and are entirely different from those obtained by traditional word embedding arithmetics (see our mentioned ablation above). We show that SD can blend concepts in a way in which each concept plays a completely different role that can be image-based (shape, texture, color), semantic (the essence of the concept), or abstract (describing a property of the concept). This is, in our view, surprising and quite remarkable. One may not agree with our conclusions, but given the magnitude of evidence we present, which was increased by the various additional experiments conducted following the reviews (OOD experiments, another text-to-image model, robustness to the loss function, etc.), we find it hard to ignore.
>
> If you found our previous response and this one to be mostly satisfactory, we kindly ask you to consider increasing your rating in recognition of the strengths and extensiveness of our evaluation, and the value of our work to the research community as the first interpretability work for text-to-image generative models.

---

> ### Comment · Reviewer_vrPv · 2023-11-21
>
> Thank you for addressing my questions.
>
> I have increased my score from 5 to 6.

---

> > ### Author Response · Authors · 2023-11-21
> >
> > Thank you for considering our rebuttal and adjusting the score! We truly appreciate your recognition and feedback.

---

### Official Review · Reviewer_nkKA · 2023-10-30

**Soundness:** 3 good
**Presentation:** 4 excellent
**Contribution:** 3 good
**Rating:** 6
**Confidence:** 5

**Summary:**

The paper ‘Hidden Language of Diffusion Models’ designs an interpretability framework for text-to-image generative models. This framework relies on learning coefficients of word-embeddings such that the reconstruction loss in diffusion models is minimized with additional constraints to ensure sparsity of concepts which are selected.  Overall, the paper provides a simple framework to decompose concepts into sub-concepts to interpret diffusion models.

**Strengths:**

- The paper is extremely well-written and easy to follow throughout. Good job on this!
- The idea about learning the coefficients of the word-embeddings while minimizing the diffusion reconstruction loss is very simple and easy-to-implement while producing good interpretability understanding. It can be a good tool to interpret diffusion models through the lens of how concept dissection works in text-to-image generative models.
- Given that there does not exist significantly good benchmarks on testing concept-decomposition and also baselines, the authors have done a satisfactory job of comparing with PEZ and other heuristic variants using BLIP-2. The ablations are also presented in depth.

**Weaknesses:**

Cons / Questions :
- While the paper provides a good interpretability framework for image generation through the lens of concepts I have some doubts on it’s downstream application. Can the authors elaborate a little bit on how Conceptor can be used for a particular downstream application (e.g., bias mitigation given that Conceptor can detect biases?)
- Can the authors elaborate if the concept decomposition is an artifact of the particular CLIP text-encoder in Stable-Diffusion? Will one get similar concept-decomposition patterns if a different text-encoder is used (e.g., T5 like in DeepFloyd)? I would imagine this to be a positive answer, but might expect different patterns, so I believe it’s important to use Conceptor to understand this phenomenon.
- I am a little curious about how much the diffusion objective plays a role in concept-decomposition. For e.g., given the objective of reconstruction, I will expect the faithfulness metric of Conceptor to be better than other methods (e.g., PEZ). However, if you use the same idea with CLIP loss (replacing the reconstruction loss in Eq. (6) with L_clip), will you get similar decomposition? And will those decompositions transfer to diffusion models?  In fact, if you use CLIP's representation for a particular token as a ground-truth with optimizing for Eq.(3), you should get a reasonable reconstruction still, which can be a cheap baseline. Did the authors run this ablation?
- How can you extend your framework to more complex concept-decomposition? The current framework generates images corresponding to single concepts, but images are usually consisting of multiple concepts. In this scenario, how can one use Conceptor to understand sub-concepts? I think this is one experiment, the paper is lacking.

**Questions:**

Refer to the previous section.

Overall, I feel that the paper is good but will like the authors to respond to the Cons/Questions.  The major question I have about this framework,(i)  is how can it be used to mitigate some of the issues in diffusion models (e.g., bias)?

---

> ### Author Response · Authors · 2023-11-16
> **Thank you for the review! (response part 1/3)**
>
> __Re. Downstream applications (with emphasis on bias mitigation):__ First, as the reviewer kindly pointed out, Conceptor enables human-understandable interpretations for diffusion models.
> Other than the useful downstream tasks (described below), we believe that interpreting diffusion models is an imperative, timely, and underexplored task on its own since these models are now being employed widely. The insights obtained by Conceptor can help both end-users and researchers to use these models responsibly while being aware of their shortcomings. Thus, we believe that interpreting generative models is, by itself, a worthy cause with much interest to our community.
>
> With that, Conceptor also enables useful downstream tasks such as concept debiasing via semantic concept editing, that we are happy to elaborate on. As mentioned in the qualitative results section (Sec. 4.1), Conceptor allows one to control each element in the decomposition by manipulating its corresponding coefficient. The magnitude of the coefficient controls the extent to which the element is manifested in the image. Fig. 3 demonstrates such manipulations. For example, one can decouple the two meanings of a dual-meaning concept (see first two rows of Fig. 3) or remove an undesired property from the concept (e.g., “nerdy” for “professor”).
>
> As mentioned in Sec. 4.3, this ability is also useful for bias mitigation. Once a bias is identified by Conceptor, the user can choose to “switch off” the biased elements by simply lowering their corresponding coefficients until an equal representation is achieved. Such examples can be found in Appendix H. For better readability, we have edited the Appendix to include the biased terms of Fig. 13, see Tab. 7. As can be seen, Conceptor enables debiasing while also maintaining the other image features and the original scene, i.e., Conceptor’s decomposition can surgically remove the biased property without harming the other features.
>
> Additionally, following the reviews, we have added an experiment showcasing another downstream task. In Appendix J, we explore Conceptor’s ability to provide insight into the representation of Out of Domain (OOD) concepts, defined by a set of image samples. As we demonstrate, our method is able to extract meaningful and profound decompositions even for completely OOD concepts (e.g., the sloth plush in Fig. 16 is linked to the concepts “plush”, “fluffy”, and “knitted”, the monster toy in Fig. 17 is linked to “woolly” due to its wool-like fur, to the character “Domo” due to the shape of its body, and to a “crab” and a “shrimp” due to its unique limbs). Kindly refer to our answer to Reviewer HDAw for more details.

---

> > ### Author Response · Authors · 2023-11-16
> > **Thank you for the review! (response part 2/3)**
> >
> > __Re. Using CLIP for alternative objective functions:__ We thank the reviewer for these very interesting questions for analysis. In the following, we address each of the suggestions separately:
> >
> > 1. **Replacing the MSE loss with a CLIP (image) similarity loss:** The implementation of this suggestion is not trivial since a CLIP loss is not applicable to the noise vectors (unlike the MSE loss). In order to facilitate the use of CLIP loss instead of MSE loss, we employed the CLIP image-to-image cosine similarity loss on the *predicted clean image* ($pred\textunderscore x_0$) at each optimization step. The results of the decompositions obtained by this experiment (on the same concepts and random seeds as Fig. 4 of the main paper), and additional technical and mathematical details on the CLIP loss used are presented in Appendix K of the revision.
> >
> >    As can be expected, the reconstructions obtained by this method fall short of the reconstructions of our full method (see Fig. 18 in the Appendix). However, importantly, the decomposition is still meaningful when replacing the MSE loss with the CLIP loss. For example, the concept “painter” still significantly relies on *the same famous artists* (e.g., “Monet”, “Picasso”), the concept “bird” still decomposes mainly into an interpolation of bird breeds, etc. We find it very reassuring that Conceptor is not sensitive to the loss function used.
> >
> > 2. **Using a specific word embedding as the ground truth for reconstruction:** Thank you for this great suggestion, which serves as a sanity check and helps us verify that our optimization indeed converges to a minimum over the objective function. Note that in the proposed setting, we are given a ground truth word embedding *from the vocabulary* $w\in \mathcal{V}$, and we are trying to optimize a *sparse linear combination over $\mathcal{V}$* to reconstruct $w$. The easiest solution that satisfies both perfect reconstruction and perfect sparsity is simply taking a “one-hot”-like vector of coefficients where $w$ is assigned the highest coefficient. This is exactly what happens when applying Conceptor with such a word embedding as ground truth. For example, for the concept “nurse”, the token “nurse” is assigned a coefficient of $1.3221e+01$, where the second highest coefficient is $1.1319e-03$ (virtually this is a one-hot vector of “nurse”). These results imply that when a simple optimal solution can be found, our MLP converges to that.
> >
> >     Importantly, unlike word embedding decomposition (which can simply be reconstructed by the embedding itself), a diffusion model gradually adds visual features from coarse to fine. This has already been established by personalization works such as NeTI [1], which shows specifically that optimizing a reconstruction token in *different timesteps* yields *entirely different results*. Therefore, each optimization step draws a different timestep where different features are added by the model, thus Conceptor’s diversity stems from the diversity of the features added in the different denoising steps, and not from the word embeddings. This point further exemplifies the strong connection between our method and the model itself, i.e.,  *the decompositions stem from the diffusion model, rather than the text encoder*. Additionally, it provides a sanity check for our decomposition, which converges appropriately. For additional empirical results, please refer to Appendix C.1 where we show that the learned pseudo-token $w^*$ can denoise *test concept images from any timestep*, and the differences between the reconstruction loss of different timesteps.
> >
> > [1] A Neural Space-Time Representation for Text-to-Image Personalization, Alaluf et al., SIGGRAPH Asia’23.

---

> > > ### Author Response · Authors · 2023-11-16
> > > **Thank you for the review! (response part 3/3)**
> > >
> > > __Re. Analysis of other text encoders:__ Thank you for this great suggestion, which helps further substantiate Conceptor’s robustness. We note that DeepFloyd text embedding optimization suffers from significant instabilities in terms of the loss and the learning rate [see this thread from the popular `diffusers` library](https://github.com/huggingface/diffusers/issues/3356#issuecomment-1578072917). The obtained gradients for the model are very low, forcing the use of very high learning rates, and the optimization is long and expensive (around 1 hour per concept requiring 24GB, *with an unconstrained token*, see [the implementation](https://github.com/oss-roettger/T5-Textual-Inversion)). To empirically substantiate the instability claim, [here are the results we got for optimizing the concept “happiness” from Fig. 4 without constraints on the word embedding](https://i.imgur.com/iTaGPzd.jpeg). This experiment is done with the same setting as Conceptor, but using unconstrained optimization of the embedding vector, which is a much more basic task. These instabilities could stem from a number of different reasons. For example, this could stem from, [DeepFloyd predicting the variance in addition to the noise](https://github.com/huggingface/diffusers/issues/3307) which is not accounted for in the current optimization, DeepFloyd operating on the pixel space and not the latent space, or due to the fact that the model is cascaded and optimized on the lowest resolution. We believe that in general, applying text embedding optimization to DeepFloyd is an interesting and non-trivial task for future research, as the behavior of the model significantly differs from LDMs. However, this deviates from the scope of our work.
> > >
> > > As an alternative, in order to address the reviewer’s question, we have applied Conceptor over another version of Stable Diffusion, SD 1.4, which uses a *different text encoder*. While SD 2.1 uses OpenCLIP-ViT/H, SD 1.4 uses CLIP ViT-L/14. Kindly note that these text encoders were trained on entirely different data (OpenCLIP is an open-source model trained on open-source data). The results are enclosed in Appendix L. As can be seen, Conceptor provides faithful decompositions, even for a model with a different text encoder, further substantiating our method’s robustness.
> > >
> > > __Re. Complex concepts containing multiple objects:__ Please note that we incorporated a subset of very complex prompts in our tests, extracted entirely from the website [Best 30 Stable Diffusion Prompts for Great Images](https://mspoweruser.com/best-stable-diffusion-prompts/). These prompts contain integrations of multiple objects/ concepts in the same scene. For example, the prompt “impression of Japanese serenity” includes multiple concepts such as pagoda, Sakura, bonsai, etc. (see Fig. 2 for example), and the prompt “marketplace colors of Marrakesh” contains an extensive variety of elements such as “spices”, “tents”, “market”, “desert”, etc. (see Tab. 8 in Appendix K, Fig. 4 of the main paper). As can be observed in our qualitative and quantitative experiments, Conceptor faithfully captures all the concepts in the prompts, even in such complex cases.
> > >
> > > We are happy to address any other questions.

---

> > > > ### Comment · Reviewer_nkKA · 2023-11-21
> > > > **Response to Authors**
> > > >
> > > > The authors have addressed my questions!  I would maintain my rating therefore!

---

> > > > > ### Author Response · Authors · 2023-11-21
> > > > >
> > > > > Thank you for considering our rebuttal and for your continued support of our work! We truly appreciate your recognition and feedback.

---

### Official Review · Reviewer_3WNH · 2023-11-01

**Soundness:** 2 fair
**Presentation:** 2 fair
**Contribution:** 2 fair
**Rating:** 6
**Confidence:** 3

**Summary:**

This paper attempts to explore the correlations between different textual concepts, by exploring how well they can help reconstruct images of a certain concept with diffusion models. The method is a variation of textual inversion, by incorporating many words from a vocabulary and learning the weights of words (instead of embeddings a new word).

**Strengths:**

The idea of finding the correlations between different textual concepts is interesting. The authors presented some interesting results, such as "snake = twisted + gecko" (Figure 2).

**Weaknesses:**

** EDIT** Thanks the author response. I've verified and indeed in the CLIP text space, the triangular similarity relationships are very noisy and seem not reflect true semantic similarity. Therefore I'd raise my rating to 6. As a relevant piece of observation, my original comments on the CLIP triangular relationships are kept as below.

====== original comment ======

My biggest concern is that it's unnecessary to use image reconstruction as a proxy to find the combination weights (Eq.3). This method can totally work in the CLIP text space only. I've tried to compute the textual similarity of the words presented in Figure 2:

Triplet: 'camel' and 'giraffe'  'cashmere'
- 'camel' vs 'giraffe': 0.834
- 'camel' vs 'cashmere': 0.774
- 'camel' vs 'giraffe' + 'cashmere': 0.872

Triplet: 'snail' and 'ladybug'  'winding'
- 'snail' vs 'ladybug': 0.768
- 'snail' vs 'winding': 0.816
- 'snail' vs 'ladybug' + 'winding': 0.855

Triplet: 'dietitian' and 'pharmacist'   'nutritious'
- 'dietitian' vs 'pharmacist': 0.878
- 'dietitian' vs 'nutritious': 0.874
- 'dietitian' vs 'pharmacist' + 'nutritious': 0.915

Triplet: 'snake' and 'twisted'  'gecko'
- 'snake' vs 'twisted': 0.869
- 'snake' vs 'gecko': 0.848
- 'snake' vs 'twisted' + 'gecko': 0.913

Triplet: 'reflections of earth' and 'sphere'    'civilization'
- 'reflections of earth' vs 'sphere': 0.761
- 'reflections of earth' vs 'civilization': 0.804
- 'reflections of earth' vs 'sphere' + 'civilization': 0.831

Triplet: 'fear' and 'scream'    'wolf'
- 'fear' vs 'scream': 0.892
- 'fear' vs 'wolf': 0.875
- 'fear' vs 'scream' + 'wolf': 0.926

We can see that for a triplet A,B,C, the similarity of A vs. (B+C) is always higher than A vs B or A vs C. That means similar semantic correlations already exist in the CLIP text embedding space. Intuitively, since CLIP text embeddings are to be aligned with image features, such similarities in the image features will propagate to the text embedding space.

Therefore, doing image reconstruction with T2I diffusion model is unnecessary. If we only mine such triplets from the CLIP text embedding space, then the contribution of this paper becomes quite small. Therefore, I suggest rejection.

**Questions:**

N/A

---

> ### Author Response · Authors · 2023-11-10
> **Thank you for the review!**
>
> We thank the reviewer for providing the feedback and especially for aiming to validate our assumptions.
>
> 1. Kindly note that our method aims to provide an interpretation of the model’s latent representation of concepts. Therefore, faithfulness is a crucial component of our method (see Sec. 4), i.e., we strive to decompose the concept into elements that can *reconstruct* it. When applying the logic suggested by the reviewer to decompose concepts, faithfulness is not guaranteed and in fact, in most cases, it does not hold.
>
>    For example, an alternative decomposition for "snail" under this logic could be "table" + "cake" since `sim("snail", "table") = 0.7749`,  `sim("snail", "cake") = 0.7847`, and `sim("snail", "table" + "cake") = 0.8110 > max{sim("snail", "table"), sim("snail", "cake")}`. However, the resulting images for the decomposition of "table" + "cake" plugged into SD provide images that are entirely different from snails, such as the two examples in [this link](https://i.imgur.com/sNtHAlX.jpeg).
>
>    Additional such examples can be found very easily, for example, one could replace "cashmere" in the decomposition for "camel" and decompose it into "giraffe" + "book" as follows: `sim("camel", "giraffe") = 0.834`,  `sim("camel", "book") = 0.7485`, `sim("camel", "giraffe"+"book") = 0.8408 > max{sim("camel", "giraffe"), sim("camel", "book")}`, see examples of SD generation for this decomposition in [this link](https://i.imgur.com/iW79uRw.jpeg). Other possibilities to replace "cashmere" in the decomposition include: "technology" (`sim("camel", "technology")=0.786`, `sim("camel", "giraffe" + "technology")=0.866 > max{0.834, 0.786}`), "cattle" (`sim("camel", "cattle")=0.832`, `sim("camel", "giraffe" + "cattle")=0.873 > max{0.834, 0.832}`) and many other words that are not semantically related to "camel" and do not produce a visual reconstruction.
>
>    Note that our method learns weighted decompositions of elements to reconstruct the concept images (see Tab. 1, Fig. 4 of the paper for example). The examples presented in Fig. 2 are actual decompositions of images generated by SD, and reconstructed by Conceptor. The image reconstruction objective is the component that guarantees a connection between the decomposition and the concept itself, which otherwise would not exist, as demonstrated in the examples above.
>
>
> 1. We note that the inequality suggested by the reviewer, i.e., `sim(A+B, C) > max{sim(A, C), sim(B, C)}` often holds for any triplet of three words. This may be a result of increased similarity with the “blurred” word obtained by averaging two concepts. To validate this empirically, we selected 10,000 random triplets of words (A, B, C) out of the SD text encoder dictionary. In 85% of the cases, the inequality holds. This indicates that the proposed inequality does not necessarily suggest a better semantic similarity, nor that A+B is a decomposition of C.
>
>
> 1. We would like to point out that the CLIP text-text similarity scores (as calculated by the reviewer using CLIP ViT-B/32) appear to give abnormally high similarities for various words that are unrelated semantically. We drew 10,000 pairs of random words from SD’s vocabulary to demonstrate this point and found that the average similarity between these random words was 0.815. These high scores are somewhat surprising, especially given that a large portion of the tokens in this dictionary are not actual words but rather punctuation marks, emojis, etc. Additionally, the scores are often unintuitive. For example, the CLIP text-text similarity between "cat" and "house" is 0.79, which is higher than the CLIP text-text similarity between “cat” and “poodle” (0.74) (to reiterate the previous point, “cat” to ”house”+“poodle” is 0.81).
>
> 1. Finally, please note that the latest version of SD (SD 2.1) employs a *different CLIP text encoder*, i.e., OpenCLIP ViT-H (see Appendix A). When examining the CLIP text-text similarities in the OpenCLIP ViT-H encoder, the phenomena described in the review do not reproduce. First, we find that the CLIP text-text similarities between the elements are *far lower* than those indicated by the reviewer (with CLIP ViT-B/32), e.g., the similarity between "reflections of earth" and "civilization" is merely 0.2817, the similarity between "snake" and "twisted" is 0.365, etc. Second, the logic indicated for CLIP ViT-B/32 is not reproduced. In 2/6 triplets mentioned by the reviewer, the sum of the two elements does not increase the similarity:
>
>    * For "snake": `sim("snake", "twisted") = 0.365`, `sim("snake", "gecko") = 0.639`, `sim("snake", "twisted" + "gecko") = 0.627 < 0.639`.
>    * For "fear": `sim("fear", "wolf") = 0.326`, `sim("fear", "scream") = 0.638`, `sim("fear", "wolf" + "scream") = 0.60 < 0.638`.
>
>    For this text encoder, the inequality holds for 65% of the random triplets as described in item 2.
>
>
> We would happily address any other questions.

---

> ### Author Response · Authors · 2023-11-19
> **Follow-up**
>
> We appreciate the diligent efforts put into reviewing our work. Like you, we believe that performing sanity checks to validate the underlying assumptions is important.
>
> May we inquire if you have had the opportunity to review our response from November 11?
>
> Thank you,
>
> The authors

---

> ### Comment · Reviewer_3WNH · 2023-11-21
> **Updated my original assessment**
>
> Thanks the author response. I've verified and indeed in the CLIP text space, the triangular similarity relationships are very noisy and seem not reflect true semantic similarity. Therefore I'd raise my rating to 6. As a relevant piece of observation, my original comments on the CLIP triangular relationships are kept.

---

> > ### Author Response · Authors · 2023-11-21
> >
> > Thank you for considering our rebuttal and adjusting the score! We truly appreciate your recognition and feedback.

---

### Official Review · Reviewer_HDAw · 2023-11-08

**Soundness:** 3 good
**Presentation:** 2 fair
**Contribution:** 2 fair
**Rating:** 6
**Confidence:** 4

**Summary:**

The paper delves into understanding the internal representations of text-to-image diffusion models, which have shown significant prowess in generating high-quality images from textual concepts. The primary challenge addressed is deciphering how these models map textual prompts to rich visual representations. The authors introduce a method, "CONCEPTOR", that decomposes an input text prompt into a set of interpretable elements. This decomposition is achieved by learning a pseudo-token, which is a sparse weighted combination of tokens from the model's vocabulary. The goal is to reconstruct the images generated for a given concept using this pseudo-token. The method facilitates single-image decomposition into tokens and semantic image manipulation.

**Strengths:**

The authors propose a novel view to interpret the internal representations of T2I diffusion model, decomposing the input text prompts into a set of prototypes. Image manipulation can be implemented by simply adjusting the coefficients of these prototypes

The method is general and flexible, as it can be applied to any T2I diffusion model without modifying the model architecture or training procedure.

The writing is good and clear. The paper also provides empirical evidence to support the effectiveness and efficiency of the method.

**Weaknesses:**

The number of concepts (prototype) are limited, which can not prove whether the proposed method is effective on large-scale concepts.

The concept decomposing can be viewed as an inner interpolation between the concepts. What if the image are out of domain? Is it possible to show some cases? Can you provided some analysis the between the proposed method and interpolation method?

**Questions:**

See the Weaknesses.

---

> ### Author Response · Authors · 2023-11-13
> **Thank you for the review!**
>
> We thank the reviewer for the positive and comprehensive feedback. We are encouraged that the reviewer found our work to be a novel, flexible, and efficient solution for diffusion model interpretability.
> In the following, we provide a response to each of the questions in the review. Additionally, the modifications in the revision are marked in red, for your convenience.
>
> __Re. Size of test dataset:__ First, kindly note that the size of our dataset (188 concepts from diverse categories) is consistent with, and even significantly bigger than that of related works that tackle concept-based interpretability and concept analysis. For example, one of the most prominent works on concept-based explainability, ACE [1], presents a quantitative evaluation of 100 random ImageNet classes. Similarly, seminal works in concept personalization such as DreamBooth [2] performed evaluation on 30 diverse concepts, etc.
>
> Importantly, note that we formed a diverse dataset including professions (to indicate biases and human-centered concepts), abstract concepts (including emotions and actions), very complex concepts (requiring hierarchical reasoning), and on top of those, we added 100 random concepts from the well-known general concept bank, ConceptNet, to ensure that our produced results are robust across a wide variety of rich and unrelated concepts. Note that ConceptNet contains a wide mixture of concepts from all categories, and is a very widely used knowledge graph. Across all these different types of concepts, Conceptor has *consistently* demonstrated an ability to produce faithful, meaningful, and robust decompositions.
>
> [1] Towards Automatic Concept-based Explanations, Ghorbani et al. NeurIPS 2019
>
> [2] DreamBooth: Fine Tuning Text-to-Image Diffusion Models for Subject-Driven Generation, Ruiz et al., CVPR 2023.
>
> __Re. Out-of-domain (OOD) images:__  Thank you for this question. While the goal of our work is to interpret the internal representations of the model (i.e., *in-domain concepts*) and images generated by the model for textual concepts, this question is indeed very interesting. Following the review, we have added a new Appendix (J) in which we examine three domains out of the DreamBooth dataset [2], with an increasing amount of distance from the training domain of SD. As can be seen, our method is able to extract meaningful and profound decompositions even for OOD concepts (e.g., the sloth plush in Fig. 16 is linked to the concepts “plush”, “fluffy”, and “knitted”, the monster toy in Fig. 17 is linked to “woolly” due to its wool-like fur, to the character “Domo” due to the shape of its body, and to a “crab” and a “shrimp” due to its unique limbs).  Moreover, our optimization results in a virtual token $w^*$ that is able to generate new OOD images at least as effectively as unrestricted optimization of the virtual token. We find these results to be extremely reassuring.
>
> __Re. Analysis between Conceptor and interpolation methods:__ Could you please clarify which interpolation method you are referring to? Given more details, we would be more than happy to conduct the appropriate analysis.
>
> We would happily address any other questions.

---

> > ### Comment · Reviewer_HDAw · 2023-11-21
> >
> > Thanks for your feedback.  The OOD experiment results are impressive.
> >
> > Regarding the interpolation, let me clarify. Semantic interpolation involves the weighted summation of the condition embeddings and generating images conditioned on the interpolated text embeddings.
> >
> > From a technical perspective, it seems that Conceptor and semantic interpolation are quite similar.

---

> ### Author Response · Authors · 2023-11-21
>
> Thank you for considering our rebuttal! We share your enthusiasm for the OOD results, and deeply value the insightful suggestion that helped enhance our understanding of Conceptor.
>
> Upon reviewing the clarification, we are still not sure we fully understand the proposed interpolation method and how it selects the condition embeddings. We derived two possible interpretations for the question which we address below, and we welcome your corrections if our understanding is not accurate.
>
> 1. Embedding interpolation as simple summation of word embeddings (i.e., summation of embeddings in the token space):
>
>    Allow us to emphasize the following:
>
>     * Recall that our method operates by learning a mapping function between each word embedding and a corresponding coefficient. In other words, our method can be viewed as a nonlinear mapping of the form: $w^* = \sum_i f(w_i) w_i$, where $f$ is a nonlinear function.
>
>        Our extensive ablations (refer to Section 4.2.1, Table 3 under "without MLP") demonstrate the critical role of this novel approach in the success of our method. Without it (i.e., learning separate coefficients) the decompositions are not faithful. This ablation highlights the non-trivial nature of assigning appropriate coefficients to word embeddings to reconstruct the learned representation, affirming that a simple summation cannot substitute for Conceptor.
>
>    * To demonstrate the important role of our learned coefficients empirically, we enclose the results of simple word embedding summation (a+b) for examples of concepts from Fig 2. The connection “reflections of earth = sphere + civilization” is only made possible by the learned coefficients. When considering only the word embeddings, [these are the resulting images](https://i.imgur.com/6ai71Ww.jpg) (which do not resemble planet Earth whatsoever). Similarly, “snake = twisted + gecko” would yield [these images without the appropriate coefficients](https://i.imgur.com/1UYOham.jpg), for the combination “snail = winding + ladybug” [these are the resulting images](https://i.imgur.com/EB4S5ya.jpg), etc. These examples demonstrate that even given the ground truth set of decomposition concepts, a simple summation cannot match our method.
>
>    We believe that the combination of the aforementioned points effectively demonstrates the powerful and non-trivial ability of Conceptor to establish semantic links between concepts through our novel method.
>
> 2. Encoding interpolation after applying the text encoder (i.e., summation of embeddings in the conditioning space):
>
>    In exploring this scenario, we observed that such interpolations result in non-meaningful images (perhaps due to OOD encodings). [Here’s an example for “ladybug” + “winding”](https://i.imgur.com/QrwUgBN.jpg), and [“twisted” + “gecko”](https://i.imgur.com/0ddljfw.jpg).
>
> We are happy to address any remaining questions, and appreciate the continued in-depth discussion.

---

> > ### Comment · Reviewer_HDAw · 2023-12-01
> >
> > Thanks for your clear reply. I have no more concern.
> >
> > I keep my suggestion, accepting this paper.

---

### Author Response · Authors · 2023-11-16
**Summary of changes in the revised manuscript**

We thank all the reviewers for providing useful and insightful feedback, and for raising interesting points for discussion. We are excited that the reviewers found our work to be novel, general, and flexible (Reviewer HDAw), to offer a tool for providing human-understandable interpretations of text-to-image diffusion models (Reviewer HDAw, Reviewer nkKA, Reviewer vrPv) to present interesting results (Reviewer 3WNH), and extensive evaluation (Reviewer HDAw, Reviewer nkKA, Reviewer vrPv), and the writing to be clear and well-motivated (Reviewer HDAw, Reviewer nkKA, Reviewer vrPv).

Our work has benefited tremendously from the feedback. Below are the main modifications to the manuscript:

1. *Out Of Domain (OOD) experiments at the request of Reviewer HDAw (Appendix J).* Conceptor is shown to generalize well to new domains and is successful in producing meaningful and interesting decompositions for OOD concepts. This supports not only the robustness of the method but also differentiates it from language model-based methods since the input is not a phrase but a set of images, a new task that we did not consider in the original manuscript.

2. *Experiments replacing the MSE loss with the CLIP image loss, following the request of Reviewer nkKA (Appendix K).* Conceptor is shown to be successful when using the *CLIP image similarity* as the reconstruction objective. This required a slight generalization of the formulation of our objective and is a strong demonstration of the method’s robustness.

3. *Robustness to the choice of the model following the request of Reviewer nkKA (Appendix L).* Conceptor is shown to work well on a different version of stable diffusion, which uses another text encoder that is trained on a different dataset.

4. *A comparison to object-centric representation methods is added at the request of Reviewer vrPv (Appendix G).* This related yet different line of work is now cited extensively (Sec. 2). Additionally, Appendix G demonstrates that the SOTA applicable method in this field, which uses stable diffusion, groups together all images from a single concept as a single token, and is not useful as a concept decomposition method. This is expected since these methods are developed for clustering samples from multiple classes.

5. *Following the request of Reviewer nkKA, we have added a table and a figure to Appendix H detailing the downstream task of debiasing.* As can be seen, the results reflect very well the removal of the biased components of each concept, while preserving the other features in the scene.

6. *Word cloud results for abstract and complex concepts are added to address a request by Reviewer vrPv (Appendix F).* In our dataset, we include abstract concepts as well as complex phrases and prompts, which were obtained from a website demonstrating high-level stable diffusion art. Such decompositions are depicted as word clouds, presenting Conceptor’s ability to identify meaningful connections for both abstract and complex concepts.


We would happily address any remaining questions by the reviewers, and continue to engage in discussion. If you found our response to be satisfactory, we kindly ask you to consider increasing your rating in recognition of the strengths and extensiveness of our evaluation, and the value of our work to the research community as the first interpretability work for text-to-image generative models.

---

### Meta-Review · Area_Chair_Vgn8 · 2023-12-09

**Metareview:**

This paper propose a new method to interpret the internal representation of text-to-image diffusion models using a set of human understandable sub-concepts. Most reviewers think the draft is well-written and the proposed method is novel and easy to implement while produce good interpretability. There were some concerns and questions on the methods, but it seems to be well addressed by the authors, where 2 reviewers increased their rating to acceptance. The authors and reviewers actively engaged in the rebuttal period, and the authors have revised the draft to address reviewers concerns and questions.

Based on above reasons, I recommend accept (poster).

**Justification For Why Not Higher Score:**

There is no accept rating (8) from any of the reviewers. Hence, recommending accept (poster).

**Justification For Why Not Lower Score:**

This paper meets the standard of ICLR paper.

---

### Decision · Program_Chairs · 2024-01-16

Accept (poster)